# On Adaptivity in Quantum Testing

**Omar Fawzi**                                                    *omar.fawzi@ens-lyon.fr*
*Univ Lyon, Inria, ENS Lyon, UCBL, LIP, F-69342, Lyon Cedex 07, France*

**Nicolas Flammarion**                                           *nicolas.flammarion@epfl.ch*
*EPFL, Lausanne, Switzerland*

**Aurélien Garivier**                                            *aurelien.garivier@ens-lyon.fr*
*Univ Lyon, ENS de Lyon, UMPA UMR 5669, F-69364 Lyon Cedex 07, France*

**Aadil Oufkir**                                                 *aadil.oufkir@ens-lyon.fr*
*Univ Lyon, Inria, ENS Lyon, UCBL, LIP, F-69342, Lyon Cedex 07, France*

**Reviewed on OpenReview:** *https://openreview.net/forum?id=Hf95zFnQ7H*

## Abstract

Can adaptive strategies outperform non-adaptive ones for quantum hypothesis selection? We exhibit problems where adaptive strategies provably reduce the number of required samples by a factor four in the worst case, and possibly more when the actual difficulty of the problem makes it possible. In addition, we exhibit specific hypotheses classes for which there is a provable polynomial separation between adaptive and non-adaptive strategies – a specificity of the quantum framework that does not appear in classical testing.

## 1 Introduction

Testing properties of quantum states is an important question in quantum learning theory which generalizes testing properties of discrete probability distributions (Nielsen & Chuang, 2002; Montanaro & de Wolf, 2013; Arunachalam & de Wolf, 2017; Anshu & Arunachalam, 2023). Indeed, a quantum state admits an intrinsic probability description given by the list of its eigenvalues. Learning completely a quantum state is expensive and requires a significant number of copies even at the presence of quantum memory (Haah et al., 2016). Since quantum resources are costly, it is crucial to design procedures to efficiently test the important properties of quantum systems. A simple way of reducing the sample complexity is to allow the testing algorithm to update its stopping rule (sequential) and/or its way of acquiring new data (adaptive) depending on the previously observed data. Here we focus on the question of the effect of sequential and adaptive strategies on the sample efficiency. This is a very active area in statistics and machine learning starting from the works of Wald (Wald, 1945) and in the learning literature under the name of bandit problems (Lattimore & Szepesvári, 2020).

Here, we consider this question for quantum testing problems. Specifically, we focus on the hypothesis selection problem using incoherent measurements, where the tester is asked to determine the hypothesis set containing the unknown quantum state $\rho$ with high probability. This problem is ubiquitous in the quantum learning theory literature, and several variants are considered: testing identity (O'Donnell & Wright, 2015; Bubeck et al., 2020; Chen et al., 2022c), testing closeness (Yu, 2020), binary hypothesis testing (Hiai & Petz, 1991), (Audenaert et al., 2007; Nussbaum & Szkoła, 2009), composite quantum hypothesis testing (Bjelaković et al., 2005). If the tester is limited to incoherent measurements, the problem is very related to classical testing problems. Indeed, on the one hand, every classical testing problem on discrete distributions can be cast into a quantum testing problem by taking diagonal quantum states corresponding to the discrete distributions. Measuring these quantum states is equivalent to sampling from the classical distributions. On the other hand, the quantum hypothesis selection problem can be seen as a bandit problem (see e.g. (Garivier

|  | Non Sequential | Sequential |
|---|---|---|
| Non adaptive | $\frac{2\log(1/\delta)}{\varepsilon^2}$ (Prop. 3.1) | $\frac{\log(1/\delta)}{2\varepsilon^2}$ (Prop. 3.3) |
| Adaptive | $\frac{2\log(1/\delta)}{\varepsilon^2}$ (Prop. 3.1) | $\frac{\log(1/\delta)}{2\varepsilon^2}$ (Prop. 3.3) |

Table 1: Copy complexity for testing whether $\rho = \sigma_1$ or $\rho = \sigma_2$ where $\varepsilon = \|\sigma_1 - \sigma_2\|_{\mathrm{tr}}$ and $\delta$ is the error probability.

|  | Non Sequential | Sequential |
|---|---|---|
| Non adaptive | $\Theta\left(\frac{d^{3/2}\log(1/\delta)}{\varepsilon^2}\right)$ (Bubeck et al., 2020) | $\mathcal{O}\left(\min\left\{\frac{d^{3/2}\log(1/\delta)}{\varepsilon^2}, \frac{d^{1/2}\log(1/\delta)}{\|\rho - \mathbb{I}/d\|_2^2}\right\}\right)$ (Prop. 3.4) |
| Adaptive | $\Theta\left(\frac{d^{3/2}\log(1/\delta)}{\varepsilon^2}\right)$ (Chen et al., 2022a) | $\mathcal{O}\left(\min\left\{\frac{d^{3/2}\log(1/\delta)}{\varepsilon^2}, \frac{d^{1/2}\log(1/\delta)}{\|\rho - \mathbb{I}/d\|_2^2}\right\}\right)$ (Prop. 3.4) |

Table 2: Copy complexity for testing whether $\rho = \frac{\mathbb{I}}{d}$ or $\left\|\rho - \frac{\mathbb{I}}{d}\right\|_{\mathrm{tr}} \geq \varepsilon$ where $d$ is the dimension of quantum states and $\delta$ is the error probability.

& Kaufmann, 2019; Lumbreras et al., 2022; Brahmachari et al., 2023)). Born's rule defines exactly the classical distribution of the reward when pulling a particular arm (performing a measurement). Note that these probability distributions are not arbitrary: they are governed by the unknown quantum state.

This connection leads to an important question: Can sequential strategies outperform non-sequential ones for some hypothesis selection problem with incoherent measurements? In other words, if the tester is allowed to choose the measurement device at a given step depending on the previous observations, would it require fewer copies of the unknown quantum state?

Note that from a practical point of view in quantum experiments, adaptive and sequential strategies can be implemented without much overhead. Thus, finding efficient adaptive/sequential strategies can lead to practical savings for quantum experiments as shown by Granade et al. (2017).

Classically, sequential strategies prove to have an advantage over non-sequential ones for instance for binary hypothesis testing problems (see (Wald, 1945)), testing continuous distributions (see (Zhao et al., 2016; Balsubramani & Ramdas, 2015)), testing identity and closeness problems with small alphabet size (see (Fawzi et al., 2021; 2022)). This speedup comes, mainly, from the fact that a sequential algorithm can make comparisons at each step and can respond earlier once it has the enough confidence. However, sequential strategies in the quantum setting have not only the capacity to choose the stopping time, but also to change the measurement devices adaptively. We expect then a larger gap between sequential and non-sequential strategies. To avoid confusion, *sequential* strategies can choose the stopping time according to the previous observations and thus they have random stopping times, while *adaptive* strategies are allowed to adapt their measurement devices at each step according to past observations. With these definitions, a strategy can be sequential and adaptive, sequential and non-adaptive, non-sequential and adaptive, or non-sequential and non-adaptive. When we don't specify whether the strategy is non-sequential or sequential (resp. non-adaptive or adaptive), it can be both and the statement remains true.

On the other hand, non-adaptive strategies have been shown to be optimal for many interesting quantum testing problems, including testing identity by (Chen et al., 2022a), purity testing and shadow tomography by (Chen et al., 2021), tomography by (Chen et al., 2022b). These works suggest that adaptive/sequential strategies cannot outperform non-adaptive non-sequential ones. The goal of the article is to show the contrary: there are some situations where sequential or adaptive strategies require fewer measurements than non-adaptive non-sequential ones.

**Contributions** When the number of hypotheses $m$ is equal to 2 and the hypotheses are simple (i.e., only one possible state), we can precisely characterize the optimal worst case complexity for non-sequential and sequential strategies. We show that sequential strategies outperform non-sequential ones by a factor 4. For the lower bounds, we show how to reduce this problem to the classical testing identity problem, then apply the lower bounds of Fawzi et al. (2022). For the sequential upper bound, we design stopping rules inspired by time uniform concentration inequalities. We refer to Table. 1 for a summary of these bounds.

|  | Non Sequential | Sequential |
|---|---|---|
| Non adaptive | $\tilde{\Theta}\left(\min\left\{\frac{md}{\varepsilon^2}, \frac{d^2}{\varepsilon^2}\right\}\right)$ (Prop. 4.4, 4.5) | $\tilde{\Theta}\left(\min\left\{\frac{md}{\varepsilon^2}, \frac{d^2}{\varepsilon^2}\right\}\right)$ (Prop. 4.4, 4.5) |
| Adaptive | $\tilde{\Theta}\left(\frac{d}{\varepsilon^2}\right)$ (Prop. 4.3, 4.9) | $\tilde{\Theta}\left(\frac{d}{\varepsilon^2}\right)$ (Prop. 4.3, 4.9) |

Table 3: Copy complexity for the hypothesis selection problem $(P)$ (4.1) where $d$ is the dimension of quantum states, $m$ is the number of hypotheses and $\varepsilon$ is the precision parameter. There is a polynomial separation between non adaptive and adaptive strategies when $\exp(\mathcal{O}(d)) \gg m \geq \Omega(d)$.

Moreover, we show that sequential algorithms can adapt to the actual difficulty for the testing mixedness and testing closeness problems. For this, we show a lower bound on the TV-distance between the probability distributions after measurement depending on the actual 1-norm between the quantum states (see Lemma 3.5). This inequality helps to reduce quantum testing to classical testing at the cost of a factor $1/\sqrt{d}$ ($d$ is the dimension of the quantum states) and can be useful for other applications. We refer to Table. 2 for a summary of these bounds.

For a number of hypotheses $m \geq 2$, we prove a separation between adaptive and non-adaptive strategies for a specific problem. The learner has the information that the unknown quantum state can be diagonalised in a basis amongst $m$ known orthonormal bases and would like to approximate it. See Def. 4.1 for a formal definition of this problem. We show that this problem can be solved by adaptive algorithms using $\mathcal{O}(d \log(m)/\varepsilon^2)$ copies of $\rho$. On the other hand, every non-adaptive algorithm solving this problem will require $\Omega(\min\{md/\log(m)\varepsilon^2, d^2/\varepsilon^2\})$ copies of $\rho$. The upper bounds follows from the shadow tomography algorithm of Huang et al. (2020). For the lower bounds, we construct an $\varepsilon$-separated family of quantum states close to the maximally mixed state ($\mathbb{I}/d$) and use it to encode a message from $[me^{\Omega(d)}]$. A learning algorithm can be used to decode this message with the same success probability. Hence, the encoder and decoder should share at least $\Omega(\log(m)+d)$ bits of information (Fano's inequality (Fano, 1961)). On the other hand, after each step, we show that the correlation between the encoder and decoder can only increase by at most $\mathcal{O}(\varepsilon^2 \log(m)/m + \varepsilon^2/d)$ bits for non-adaptive strategies and it can only increase by at most $\mathcal{O}(\varepsilon^2)$ bits for adaptive strategies. We obtain an improvement by a factor $d$ or $m/\log(m)$ for non-adaptive strategies by exploiting the randomness in the construction and the independence of the observations at different steps. We refer to Table 3 for a summary of these bounds.

**Related work** Quantum testing identity using entangled measurements is well understood (O'Donnell & Wright, 2015; Bădescu et al., 2019): it is known that $\Theta(d/\varepsilon^2)$ copies are necessary and sufficient. For incoherent measurements, it starts with the work of Bubeck et al. (2020) where we have two different lower bounds for testing mixedness problem using independent adaptive and non-adaptive measurements. This result is generalized for general testing identity to some quantum state $\sigma$ by Chen et al. (2022c). Recently Chen et al. (2022a) show that adaptive algorithms cannot significantly outperform non-adaptive ones neither for testing mixedness nor testing identity.

If entangled measurements are allowed, the quantum hypothesis selection problem can be solved using $\text{poly}(\log(m))$ copies of $\rho$ (see (Bădescu & O'Donnell, 2021)). This poly-logarithmic complexity in $m$ can be explained by the fact that $\rho^{\otimes N}$ can be reused after measurement. In contrast, this is not possible using incoherent measurements for which the state collapses after performing the measurement. In general, the quantum hypothesis selection problem, where each hypothesis contains only one quantum state, is highly related to the shadow tomography problem where the learner is asked to uniformly approximate the expected values $\{\text{tr}(\rho O_i)\}_{i \in [m]}$ of $m$ known observables $\{O_i\}_{i \in [m]}$ by measuring the unknown quantum state $\rho$. A popular algorithm for the shadow tomography problem is given by Huang et al. (2020) and uses at most $\mathcal{O}(\log(m)d/\varepsilon^2)$ non-sequential non-adaptive incoherent measurements. On the other hand, independent adaptive strategies are shown to be useless for shadow tomography (and purity testing) by Chen et al. (2021).

Moreover, sequential adaptive strategies have been used by Li et al. (2022b) (see (Li et al., 2022a) for quantum channel discrimination) to achieve the optimal rates given by the quantum relative entropy for both type I and type II errors at the same time for binary hypothesis testing problem using entangled measurements.

Adaptive strategies have been considered for testing quantum channels in (Harrow et al., 2010; Pirandola et al., 2019; Salek et al., 2022). In particular, Harrow et al. (2010) and Salek et al. (2022) provide examples for which adaptive strategies outperform non-adaptive ones for testing quantum channels. We note that for channels, one has the possibility to adapt the *input* of the channel to the previous observations, but this is not the case for testing states. As such, it is more challenging to find a separation between adaptive and non-adaptive strategies for testing quantum states than it is for channels.

For the tomography problem, Chen et al. (2022b) shows that adaptive independent strategies cannot beat non-sequential non-adaptive ones and thus need at least $\Omega(d^3/\varepsilon^2)$ copies to learn the quantum state $\rho$. However, it is unclear whether adaptivity can help for learning restricted families of states such as graph states (Ouyang & Tomamichel, 2022). On the other hand, sequential strategies were used for online state tomography by Kueng & Ferrie (2015); Youssry et al. (2019); Stricker et al. (2022); Rambach et al. (2022). Note that the word "online learning" is usually used to refer to a learning task where the properties or the state we want to estimate change on the fly. The problems we consider here are not of this type: the task is fixed in advance. For this reason we use the words adaptive and sequential instead of online.

Finally, other works consider testing properties of quantum states/distributions with different access models. For instance, (Acharya et al., 2020) studies the copy complexity of estimating entropies of a quantum state, (Gilyén & Li, 2019) studies testing closeness between unknown distributions with coherent access and (Belovs, 2019) studies the quantum query complexity of discriminating two probability distributions encoded by quantum oracles.

## 2 Preliminaries

Throughout the paper, $d$ is the dimension of the quantum states. A quantum state is a positive semi-definite Hermitian matrix of trace 1. We use the bra-ket notation: a column vector is denoted $|\phi\rangle$ and its adjoint is denoted $\langle\phi| = |\phi\rangle^\dagger$. With this notation, $\langle\phi|\psi\rangle$ is the dot product of the vectors $\phi$ and $\psi$ and, for a unit vector $|\phi\rangle$, $|\phi\rangle\langle\phi|$ is the rank-1 projector on the space spanned by the vector $\phi$. The canonical basis $\{e_i\}_{i\in[d]}$ is denoted $\{|i\rangle\}_{i\in[d]} := \{|e_i\rangle\}_{i\in[d]}$.

We define the trace norm or the 1-norm of a matrix $M$ as $\|M\|_{\mathrm{tr}} = \frac{1}{2}\mathrm{tr}\left(\sqrt{M^\dagger M}\right)$ and the 2-norm as $\|M\|_2 = \sqrt{\mathrm{tr}\left(M^\dagger M\right)}$. An observable is a Hermitian matrix $O$ satisfying $O \succcurlyeq 0$ and $\mathbb{I} - O \succcurlyeq 0$ where $\mathbb{I}$ is the identity matrix.

Given two quantum states $\rho$ and $\sigma$, we can compare them using the quantum relative entropy defined as:

$$D(\rho\|\sigma) = \mathrm{tr}(\rho(\log(\rho) - \log(\sigma)))$$

or the quantum Chernoff divergence defined as:

$$C(\rho, \sigma) = -\log\left(\inf_{0 \le s \le 1}\mathrm{tr}(\rho^{1-s}\sigma^s)\right).$$

The total variation (TV) distance between two probability distributions $P$ and $Q$ on $[d]$ is defined as:

$$\mathrm{TV}(P, Q) = \frac{1}{2}\sum_{i=1}^{d}|P_i - Q_i|$$

and the Kullback-Leibler (KL) divergence is defined as:

$$\mathrm{KL}(P\|Q) = \sum_{i=1}^{d}P_i\log\left(\frac{P_i}{Q_i}\right).$$

Finally, for two numbers $p, q \in [0, 1]$, we denote $\mathrm{KL}(p\|q) = \mathrm{KL}(\mathrm{Ber}(p)\|\mathrm{Ber}(q))$.

All the problems discussed in this article are special cases of the general hypothesis selection problem. Given an unknown quantum state $\rho \in \mathbb{C}^{d\times d}$ and $m$ hypothesis classes $\{H_i\}_{i\in[m]}$, the learner is asked to find one of the hypothesis classes containing $\rho$ with high probability. Formally:

**Definition 2.1** (Quantum hypothesis selection)**.** Let $\rho \in \mathbb{C}^{d \times d}$ be an unknown quantum state. Let $\{H_i\}_{i \in [m]}$ be $m$ hypothesis classes. We have the promise that at least one of the following assertions is satisfied:

$$\rho \in H_1, \rho \in H_2, \ldots, \rho \in H_m .$$

An algorithm $\mathcal{A}$ is $\delta$-correct for this problem if it verifies the following property:

$$\forall i \in [m] : \rho \notin H_i \implies \mathbb{P}(\mathcal{A} = i) \leq \delta .$$

The difference between quantum and classical testing is that in the quantum case we have the possibility to choose a measurement (given by positive operators summing to the identity). If the quantum states are restricted to be diagonal, we may assume the measurement is always the same and so the problem becomes a classical testing problem (see Lemma 3.2).

The quantum state $\rho$ is unknown, but the learner can extract classical information from it by performing a measurement. The way the unknown quantum state $\rho$ is measured is important and can lead to different results about the number of copies needed for this task.

**Definition 2.2.** A measurement is defined by a POVM (positive operator-valued measure) with a finite number of elements: this is a set of positive semi-definite matrices $\mathcal{M} = \{M_i\}_{i \in \mathcal{X}}$ acting on a Hilbert space $\mathcal{H}$ and satisfying $\sum_{i \in \mathcal{X}} M_i = \mathbb{I}_{\mathcal{H}}$. Each element $M_i$ in the POVM $\mathcal{M}$ is associated with the outcome $i \in \mathcal{X}$. The tuple $\{\text{tr}(\rho M_i)\}_{i \in \mathcal{X}}$ is non-negative and sums to 1: it thus defines a probability denoted by $\rho(\mathcal{M})$. Born's rule (Born, 1926) says that the probability that the measurement on a quantum state $\rho$ using the POVM $\mathcal{M}$ will output $i$ is exactly $\text{tr}(\rho M_i)$.

We distinguish between two types of measurements depending on the considered Hilbert space:

**Definition 2.3** (Entangled measurement)**.** An entangled measurement is given by a POVM on the Hilbert space $\mathcal{H} = (\mathbb{C}^d)^{\otimes N}$, where $N$ is the number of copies available of the quantum state $\rho$. We can measure the whole state $\rho^{\otimes N}$ at once. An interesting POVM related to the observable $O$ on $\mathbb{C}^d$ is given by $\mathcal{M}(O) = \{M_k\}_{0 \leq k \leq N}$ where $M_k = \sum_{x \in \{0,1\}^N, |x|=k} O^{x_1} \otimes \cdots \otimes O^{x_N}$. Measuring $\rho^{\otimes N}$ with the POVM $\mathcal{M}(O)$ outputs a sample from the binomial distribution $\text{Bin}(n, \text{tr}(\rho O))$.

**Definition 2.4** (Incoherent measurement)**.** An incoherent measurement is given by a sequence of POVMs $\{\mathcal{M}_t\}_{t \in [N]}$, each of them acts on the Hilbert space $\mathcal{H} = \mathbb{C}^d$. In this case, we measure at step $t$ the quantum state $\rho$ using the POVM $\mathcal{M}_t$. For instance, for an observable $O$, measuring $\rho$ with the POVM $\mathcal{M}(O) = (\mathbb{I} - O, O)$ outputs a sample from the Bernoulli distribution $\text{Ber}(\text{tr}(\rho O))$.

In this article, we focus on algorithms using incoherent measurements. In this case, we can distinguish between two four types of strategies depending whether the number of measurements and the POVMs $\{\mathcal{M}_t\}_t$ are fixed in advance or not.

**Definition 2.5** (Adaptive strategies)**.** A strategy is called *non-adaptive* when the POVMs $\{\mathcal{M}_t\}_t$ are fixed in advance (i.e., do not depend on the outcomes of the previous measurements). When $\mathcal{M}_t$ can be chosen depending on the results of the previous measurements with the $(\mathcal{M}_s)_{s<t}$, we call it an *adaptive* strategy.

**Definition 2.6** (Sequential strategies)**.** If the number of measurements is not fixed beforehand and can be chosen as a function of the previous measurement outcomes, the strategy is called *sequential* and has a random stopping time $N$. In this case, the *expected copy complexity* of the procedure is $\mathbb{E}(N)$. Otherwise, the strategy has a fixed number of measurements $N$ and is called *non-sequential*.

The goal of this article is to assess the potential improvement of sequential/adaptive algorithms over non-adaptive non-sequential ones.

## 3 Sequential improvement for problems involving two hypotheses

In this section, we focus on sequential algorithms for problems having only two hypotheses ($m = 2$), which can be simple or not. The results of this Section are true for either adaptive and non-adaptive settings.

### 3.1 Provable constant improvement of sequential strategies

The simplest case for hypothesis selection problem with $m = 2$ corresponds to hypothesis sets containing only one known quantum state. Formally, the learner would like to distinguish two hypothesis: $H_1 = \{\sigma_1\}$ and $H_2 = \{\sigma_2\}$. We want to characterize the exact number of copies the learner needs to solve this problem using sequential and non-sequential independent measurements.

#### 3.1.1 Non-sequential strategies

The tester knows the quantum states $\sigma_1$ and $\sigma_2$ and can hence calculate the actual 1-norm between them, denoted by $\varepsilon = \|\sigma_1 - \sigma_2\|_{\mathrm{tr}}$. The optimal POVM to distinguish between $\sigma_1$ and $\sigma_2$ is thus given by $\mathcal{M} = (\mathbb{I} - O, O)$ (Holevo-Helstrom theorem, see (Watrous, 2018)) where $0 \preccurlyeq O \preccurlyeq \mathbb{I}$ satisfies

$$\varepsilon = \|\sigma_1 - \sigma_2\|_{\mathrm{tr}} = \mathrm{tr}((\sigma_1 - \sigma_2)O) . \tag{1}$$

Let $X_1, \ldots, X_N$ be the outcomes of measuring $\rho$ by the POVM $\mathcal{M}$. By Born's rule, they follow the Bernoulli distribution of parameter $\mathrm{tr}(\rho O)$. Let S be the statistic given by the difference between the empirical mean and the actual mean under $H_2$: $S = \frac{1}{N} \sum_{i=1}^{N} X_i - \mathrm{tr}(\sigma_2 O)$. Its expected value is $\mathrm{tr}((\rho - \sigma_2)O)$ which is $\varepsilon$ under $H_1$ and 0 under $H_2$. The learner can measure $\rho$ a sufficient number of times, compare the statistic $S$ with $\varepsilon/2$ and decide accordingly: If $S \geq \varepsilon/2$ it accepts $H_1$, otherwise it accepts $H_2$. Following the Chernoff-Hoeffding inequality (Hoeffding, 1963), a sufficient number of measurement for the learner to be $\delta$-correct is

$$\max\left\{ \frac{\log(1/\delta)}{\mathrm{KL}(\mathrm{tr}((\sigma_1 + \sigma_2)O)/2\|\mathrm{tr}(\sigma_1 O))}, \frac{\log(1/\delta)}{\mathrm{KL}(\mathrm{tr}((\sigma_1 + \sigma_2)O)/2\|\mathrm{tr}(\sigma_2 O))} \right\} \leq \frac{2\log(1/\delta)}{\varepsilon^2} .$$

The latter inequality follows from Pinsker's inequality (Fedotov et al., 2003). Note that this previous upper bound is optimal in the worst case setting where we fix $\varepsilon$ and take the infimum over all $\sigma_1$ and $\sigma_2$ satisfying $\|\sigma_1 - \sigma_2\|_{\mathrm{tr}} = \varepsilon$. This first result is summarized in the following proposition:

**Proposition 3.1.** *There is a non-sequential non-adaptive algorithm for testing $H_1 : \rho = \sigma_1$ vs $H_2 : \rho = \sigma_2$ using a number of measurements*

$$N \leq \frac{2\log(1/\delta)}{\varepsilon^2} .$$

*Moreover, there exists two quantum states $\sigma_1$ and $\sigma_2$ satisfying $\|\sigma_1 - \sigma_2\|_{\mathrm{tr}} = \varepsilon$ so that every non-sequential adaptive algorithm distinguishing between $H_1 : \rho = \sigma_1$ and $H_2 : \rho = \sigma_2$ needs a number of measurements satisfying*

$$\liminf_{\delta \to 0} \frac{N}{\log(1/\delta)} \geq \max\left\{ \frac{1}{\mathrm{KL}(1/2 + \alpha\varepsilon\|1/2)}, \frac{1}{\mathrm{KL}(1/2 - \beta\varepsilon\|1/2)} \right\} \underset{\varepsilon \to 0}{\sim} \frac{2}{\varepsilon^2} ,$$

*where $\alpha \in (0, 1)$ and $\beta \in (0, 1)$ are defined such that $\mathrm{KL}(1/2 + \alpha\varepsilon\|1/2) = \mathrm{KL}(1/2 + \alpha\varepsilon\|1/2 + \varepsilon)$ and $\mathrm{KL}(1/2 - \beta\varepsilon\|1/2) = \mathrm{KL}(1/2 - \beta\varepsilon\|1/2 - \varepsilon)$.*

For the lower bound, construct two quantum states, $\sigma_1 = \mathbb{I}_2/2$ and $\sigma_2$ an $\varepsilon$ perturbation of it. For each POVM, we show that the optimal sample complexity is at least $\frac{2\log(1/\delta)}{\varepsilon^2}$ (when $\varepsilon \to 0$), with equality iff the POVM is the optimal one defined in Eq. (1). This reduction can be proven using the following lemma on measurements of diagonal quantum states.

**Lemma 3.2.** *Let $\mathcal{D}_1$ and $\mathcal{D}_2$ be two discrete distributions and $\rho_1$ and $\rho_2$ their corresponding diagonal quantum states. Let $\mathcal{M}$ be a POVM. Measuring the quantum state $\rho_1$ (resp. $\rho_2$) with the POVM $\mathcal{M}$ can be seen as post-processing (independent of the quantum states) of samples from the distribution $\mathcal{D}_1$ (resp. $\mathcal{D}_2$).*

Hence, for each POVM, measuring the constructed quantum states $\sigma_1$ and $\sigma_2$ is a post-processing of samples from $\mathcal{D}_1 = \mathrm{Ber}(1/2)$ and $\mathcal{D}_2 = \mathrm{Ber}(1/2 + \varepsilon)$. Note that this reduction (and the lower bound) works even for entangled strategies. Once the reduction to classical testing identity is done, we can invoke the lower bound of (Fawzi et al., 2022). The proof is deferred to App. A.1.

### 3.1.2 Sequential strategies

If we allow the tester to adapt the measurements and choose its stopping time according to previous observations, it can outperform (in expectation) every non-sequential algorithms by a factor 4. Precisely, it can be proven that an expected number of measurements asymptotically equivalent to $\frac{\log(1/\delta)}{2\varepsilon^2}$ is sufficient to distinguish between $H_1 : \rho = \sigma_1$ and $H_2 : \rho = \sigma_2$ with probability at least $1 - \delta$. We use again the optimal POVM $\mathcal{M}$ defined in Eq. (1) to distinguish between $\sigma_1$ and $\sigma_2$. Let $X_1, \ldots, X_t \sim \text{Ber}(\text{tr}(\rho O))$ the outcomes of measuring $\rho$ by the POVM $\mathcal{M}$. Let $S_t = \frac{1}{t} \sum_{i=1}^{t} X_i$ the empirical mean until the time $t$. Contrary to the algorithm described in the previous subsection, a sequential algorithm can make comparisons at each time $t$ until the tester is confident enough to answer the correct answer $H_1$ or $H_2$. Under $H_1$, the statistic $S_t$ has an expected value $\text{tr}(\sigma_1 O)$. On the other hand, under $H_1$, the statistic $S_t$ has an expected value $\text{tr}(\sigma_2 O)$. These expected values are known to the tester, so it can compare at each time the statistic $S_t$ with two thresholds: $\text{tr}(\sigma_1 O) - \phi(\delta, t)$ and $\text{tr}(\sigma_2 O) + \phi(\delta, t)$ where $\phi(\delta, t)^2 = \frac{1}{2t} \log\left(\frac{2t(t+1)}{\delta}\right)$. If $S_t \leq \text{tr}(\sigma_1 O) - \phi(\delta, t)$, the tester can answer $H_2$ confidently. Similarly, it would answer $H_1$ if $S_t \geq \text{tr}(\sigma_2 O) + \phi(\delta, t)$. However if none of these inequalities is verified it does not answer and makes a new measurement, and so forth until the regions defined by the thresholds coincide. The idea of comparing the statistic with time dependent thresholds has been previously used for classical sequential testing by Balsubramani & Ramdas (2015); Fawzi et al. (2021; 2022). In these latter articles, it is proven that in expectation this algorithm outperform the non sequential one by a factor 4. We adapt their result to the quantum setting in the following proposition.

**Proposition 3.3.** *There is a sequential non-adaptive algorithm for testing $H_1 : \rho = \sigma_1$ vs $H_2 : \rho = \sigma_2$ using an expected number of measurements:*

$$\mathbb{E}(N) \leq \frac{\log(1/\delta)}{2\varepsilon^2} + \frac{\log(1/\delta)^{2/3} + 2\log(1/\delta)^{1/3} + \log(\log(1/\delta)/2\varepsilon^2) + 1}{\varepsilon^2} \ .$$

*Moreover, there are two quantum states $\sigma_1$ and $\sigma_2$ satisfying $\|\sigma_1 - \sigma_2\|_{\text{tr}} = \varepsilon$ so that every sequential adaptive algorithm distinguishing between $H_1 : \rho = \sigma_1$ and $H_2 : \rho = \sigma_2$ needs a number of measurements satisfying:*

$$\mathbb{E}(N) \geq \frac{\log(1/\delta)}{\min\{\text{KL}(1/2 \pm \varepsilon \| 1/2)\}} \underset{\varepsilon \to 0}{\sim} \frac{1}{2\varepsilon^2} \ .$$

Note that the expected stopping time is the most natural figure of merit for the sample complexity of sequential algorithms. Moreover, using the same analysis for the algorithm, one can also obtain similar bounds on the number of measurements with high probability.

Observe that the upper bound admits the asymptotic limit $\limsup_{\delta \to 0} \frac{\mathbb{E}(N)}{\log(1/\delta)} \leq \frac{1}{2\varepsilon^2}$ hence $\frac{\log(1/\delta)}{2\varepsilon^2}$ is asymptotically the worst-case optimal complexity of discriminating between two quantum states at the limit $\delta, \varepsilon \to 0$. The correctness of the algorithm presented here is proved using the following time uniform concentration inequality which is an application of union bound and Hoeffding's inequality (Hoeffding, 1963):

$$\mathbb{P}\left(\exists t \geq 1 : |S_t - \mathbb{E}(S_t)| > \phi(\delta, t)\right) \leq \delta$$

where $\phi(\delta, t) = \sqrt{\frac{1}{2t} \log\left(\frac{2t(t+1)}{\delta}\right)}$.

The lower bound follows from the previous proof's reduction and the lower bound on the expected number of samples for testing uniform using sequential algorithms: $\text{Ber}(1/2)$ vs $\text{Ber}(1/2 \pm \varepsilon)$ (see (Fawzi et al., 2022)). The detailed proof can be found in App. A.1.

Note that Li et al. (2022b) have also established an advantage of sequential adaptive strategies over non-adaptive non-sequential ones in terms of the error exponents. The type I error is the probability that the testing algorithm answers the hypothesis $H_1$ while the hypothesis $H_0$ is the correct one while the type II error is the probability that the testing algorithm answers the hypothesis $H_0$ while the hypothesis $H_1$ is the correct one:

$$\alpha_N = \mathbb{P}_{H_0}(\mathcal{A}_N = 1) \quad \text{and} \quad \beta_N = \mathbb{P}_{H_1}(\mathcal{A}_N = 0)$$

where $N$ is the number of copies used. The error exponents (rates) are then given by

$$R_0 = \lim_{N \to \infty} \frac{-\log(\alpha_N)}{N} \quad \text{and} \quad R_1 = \lim_{N \to \infty} \frac{-\log(\beta_N)}{N} .$$

Concretely, Li et al. (2022b) show that adaptive sequential strategies can achieve the best rates (at the same time) given by the quantum relative entropy between two states for both type I and II errors. On the other hand, it is known that non sequential non adaptive strategies can only achieve the quantum Chernoff rate exponent when the error probabilities are equal (Nussbaum & Szkoła, 2009; Audenaert et al., 2007). For the particular states $\sigma_1 = \frac{\mathbb{I}_2}{2}$ and $\sigma_2 = \text{diag}(\frac{1}{2} + \varepsilon, \frac{1}{2} - \varepsilon)$, we can show that the quantum relative entropy and the quantum Chernoff divergence between $\sigma_1$ and $\sigma_2$ are asymptotically equivalent to:

$$D(\sigma_1 \| \sigma_2), D(\sigma_2 \| \sigma_1) \underset{\varepsilon \to 0}{\sim} 2\varepsilon^2 \quad \text{and} \quad C(\sigma_1, \sigma_2) \underset{\varepsilon \to 0}{\sim} \frac{\varepsilon^2}{2} .$$

Therefore, we can recover the factor 4 improvement by comparing the quantum relative entropy and the quantum Chernoff divergence:

$$\frac{D(\sigma_1 \| \sigma_2)}{C(\sigma_1, \sigma_2)}, \frac{D(\sigma_2 \| \sigma_1)}{C(\sigma_1, \sigma_2)} \underset{\varepsilon \to 0}{\sim} \frac{2\varepsilon^2}{\frac{\varepsilon^2}{2}} = 4 .$$

We refer to App. A.1.3 for the detailed computations.

## 3.2 Sequential strategies adapt on the actual difficulty of the problem without prior knowledge

In this section, we change the previous setting by letting the second hypothesis be multiple. Precisely, we consider the problem of testing identity with $H_1 = \{\mathbb{I}/d\}$ and $H_2 = \{\rho : \|\rho - \mathbb{I}/d\|_{\text{tr}} \geq \varepsilon\}$ where $\varepsilon$ is a positive parameter. (Chen et al., 2022a) has proved that the optimal non-sequential adaptive copy complexity is $\Theta(d^{3/2}/\varepsilon^2)$. We show that while non-sequential adaptive algorithms cannot improve the copy complexity, sequential non-adaptive algorithms can be used to adapt to the actual difficulty of the problem. Mainly we show the following result:

**Proposition 3.4.** *There is a sequential non adaptive algorithm for testing identity problem using a number of measurements satisfying:*

$$\mathbb{E}(N) = \mathcal{O}\left(\min\left\{\frac{d^{3/2}\log(1/\delta)}{\varepsilon^2}, \frac{d^{1/2}\log(1/\delta)}{\|\rho - \mathbb{I}/d\|_2^2}\right\}\right) .$$

In particular, the expected copy complexity can be reduced to $\mathcal{O}(rd^{1/2}\log(1/\delta))$ if the quantum state $\rho$ has low rank $r \leq d/2$ or $\mathcal{O}\left(\frac{rd^{1/2}\log(1/\delta)}{\|\rho - \mathbb{I}/d\|_{\text{tr}}^2}\right)$ if the trace-less matrix $\rho - \mathbb{I}/d$ has low rank $r$ even if the algorithm does not have any information about these ranks (see App. A.2). The algorithm uses random measurements and a time-dependent stopping rule. Since we have already sequential algorithms for the classical testing identity problem, it is sufficient to show how to reduce the quantum problem to the classical one. For a POVM $\mathcal{M}$ and a quantum state $\rho$, let $\rho(\mathcal{M})$ denotes the classical probability distribution $\{\text{tr}(\rho M_i)\}_i$. The following lemma captures the main ingredient of the reduction:

**Lemma 3.5.** *For all $\delta > 0$, let $l = \frac{1}{4}\log(2/\delta)$ and $U^1, U^2, \dots, U^l \in \mathbb{C}^{d \times d}$ be Haar-random unitary matrices of columns $\{|U_i^j\rangle\}_{1 \leq i \leq d, 1 \leq j \leq l}$, $\mathcal{M} = \{\frac{1}{l}|U_i^j\rangle\langle U_i^j|\}_{i,j}$ is a POVM and for all quantum states $\rho$ and $\sigma$ we have with a probability at least $1 - \delta$:*

$$\text{TV}(\rho(\mathcal{M}), \sigma(\mathcal{M})) \geq \frac{\|\rho - \sigma\|_2}{20} \geq \frac{\|\rho - \sigma\|_{\text{tr}}}{20\sqrt{r}} ,$$

*where $r$ is the rank of $(\rho - \sigma)$.*

It is, in general, difficult to compute the expected value of the 1-norm under Haar measure, but the 2 and 4-norms can be computed exactly with the Weingarten calculus (see Lemma D.1 and Lemma D.2), so we

use Hölder's inequality to lower bound the 1-norm by an expression involving the 2 and 4-norms. Moreover, this will only give a lower bound in expectation, so we sample more Haar-distributed unitaries and construct a new measurement device by concatenating a fraction of the columns of each unitary. Then, we need to show that the TV-distance is Lipschitz to be able to apply a concentration inequality for functions of Haar-distributed unitaries (Theorem D.3). This is done by carefully applying the Cauchy Schwarz inequality. The complete proof can be found at App. A.2. Sen (2006) proved a slightly weaker (by a logarithmic factor) lower bound on the TV distance using a POVM constructed with Gaussian random variables. Also, a similar lower bound (in expectation) can be found in (Matthews et al., 2009) where the authors analyze the uniform POVM and a POVM defined by a spherical 4-designs. However, for our reduction, it is important to minimize the number of outcomes of the POVM.

Lemma 3.5 gives a POVM for which our problem reduces to testing identity: $P = U_n$ vs $\text{TV}(P, U_n) \geq \frac{\varepsilon}{20\sqrt{d}}$ with high probability, where $n = \frac{1}{4}d\log(2/\delta)$ and $P = \mathcal{M}(\rho)$. Under the alternative hypothesis $H_2$, the TV distance between $P$ and $U_n$ can be lower bounded by $\text{TV}(P, U_n) \geq \frac{1}{20}\|\rho - \mathbb{I}/d\|_2$. Therefore we can apply the sequential classical testing uniform result of (Fawzi et al., 2022) to obtain a copy complexity

$$\mathcal{O}\left(\frac{d^{3/2}\log(1/\delta)}{\max\{\varepsilon^2, d\|\rho - \mathbb{I}/d\|_2^2\}}\right).$$

A matching lower bound can be obtained in the worst case setting where we are interested only in the parameters $d$, $\varepsilon$ and $\|\rho - \mathbb{I}/d\|_{\text{tr}}$. This can be done using Markov's inequality to transform the algorithm to a deterministic-time one then invoking the lower bound of (Chen et al., 2022a): Any adaptive algorithm for testing identity would require $\Omega(d^{3/2}/\varepsilon^2)$ copies of $\rho$.

Note that, using Lemma 3.5 and the sequential tester of (Fawzi et al., 2022), we obtain the same copy complexity for testing closeness (i.e., testing $\rho = \sigma$ vs $\|\rho - \sigma\|_{\text{tr}} \geq \varepsilon$ where we can measure the unknown quantum states $\rho$ and $\sigma$) as for testing identity. This is different from the classical case where testing identity (Diakonikolas et al., 2017) can be done with much less copies than testing closeness (Diakonikolas et al., 2020).

## 4 Provable separation between adaptive and non-adaptive strategies

In this section, we fix the error probability to $\delta = 1/3$. We construct a problem for which we have a separation between adaptive and non-adaptive algorithms.

**Definition 4.1.** [Hypothesis selection problem $(P)$] Let $\{\sigma_1, \ldots, \sigma_m\}$ be a set of $\varepsilon$-separated known quantum states. The unknown quantum state $\rho$ is $\varepsilon/3$-close to (at most) one of the quantum states $\sigma_{i^\star} \in \{\sigma_1, \ldots, \sigma_m\}$ and has the same diagonalisation basis than $\sigma_{i^\star}$. We aim to learn the quantum state $\rho$ to within $\varepsilon/10$ with high probability. Formally, the goal is to design an algorithm that measures a number of copies of $\rho$ and returns a quantum state $\tilde{\rho}$ (an $\varepsilon/10$-approximation of $\rho$) such that with probability (the randomness comes from the measurements and possibly the algorithm) at least $1 - \delta$:

$$\|\tilde{\rho} - \rho\|_{\text{tr}} \leq \frac{\varepsilon}{10}.$$

The problem described above is not a hypothesis selection problem in the strict sense of the term. However it is equivalent to the following hypothesis selection problem which has the same order of copy complexity. For $i \in [m]$, let $\sigma_i = \sum_k \lambda_k |\phi_k^i\rangle\langle\phi_k^i|$ and $\{\sigma_{i,j}\}_{j\in[M]}$ an $\varepsilon/10$-covering of the set $\{\rho = \sum_k \mu_k |\phi_k^i\rangle\langle\phi_k^i| : \text{TV}(\lambda, \mu) \leq \varepsilon/3\}$. Our problem is equivalent to the hypothesis selection problem for $\{H_{i,j} = \{B(\sigma_{i,j}, \varepsilon/10)\} \cap \{\rho : \rho\sigma_{i,j} = \sigma_{i,j}\rho\}\}_{i\in[m],j\in[M]}$. For simplicity, we use the first formulation of the problem and refer to it as $(P)$.

### 4.1 Upper bound

In this section, we present an adaptive algorithm for the problem $(P)$ achieving a copy complexity strictly less than the lower bound which holds for all non-adaptive algorithms. The first step is to determine with high probability the closest quantum state $\sigma_{i^\star}$ to $\rho$, then it remains to approximate $\rho$ by measuring it in its basis of diagonalization.

---

**Algorithm 1** Hypothesis selection problem ($P$).

---

**Input:** $N = \mathcal{O}(d \log(m/\delta)/\varepsilon^2)$ incoherent measurements on $\rho$ and $m$ quantum states $\sigma_1, \ldots, \sigma_m$.

**Output:** Two quantum states $\sigma_{i^\star}$ and $\tilde{\rho}$ satisfying with a probability at least $1 - \delta$: $\|\sigma_{i^\star} - \rho\|_{\mathrm{tr}} \le \varepsilon/3$ and $\|\tilde{\rho} - \rho\|_{\mathrm{tr}} \le \varepsilon/10$ .

For all $i \ne j \in [m]$, let $O_{i,j}$ an observable satisfying $\|\sigma_i - \sigma_j\|_{\mathrm{tr}} = \mathrm{tr}\, O_{i,j}(\sigma_i - \sigma_j)$.

For all $i \ne j \in [m]$, let $\mu_{i,j}$ an $\varepsilon/10$ approximation of $\mathrm{tr}(\rho O_{i,j})$ given by classical shadow tomography of (Huang et al., 2020).

Let $k^\star = \mathrm{argmin}_l \max_{i,j} \mu_{i,j} - \mathrm{tr}(\sigma_l O_{i,j})$.

Let $\mathcal{M} = \{|\phi_i\rangle\langle\phi_i|\}_{i \in [d]}$ the POVM corresponding to the basis of diagonalisation of $\sigma_{k^\star}$.

Measure $\rho$ independently $M = 200 \log(2^{d+2}/\delta)/\varepsilon^2$ times using the POVM $\mathcal{M}$ and denote the outcomes $\{E_i\}_{1 \le i \le M}$.

**return** $\tilde{\rho} = \sum_{i \in [d]} \left( \frac{\sum_{j \in [M]} \mathbf{1}_{E_j = i}}{M} \right) |\phi_i\rangle\langle\phi_i|$.

---

### 4.1.1 Adaptive strategies.

For all $i \ne j \in [m]$, let $O_{i,j}$ an observable satisfying $\|\sigma_i - \sigma_j\|_{\mathrm{tr}} = \mathrm{tr}\, O_{i,j}(\sigma_i - \sigma_j)$. In Sec. 3.1, we have seen that such observable $O_{i,j}$ can be used to distinguish between $\rho = \sigma_i$ and $\rho = \sigma_j$ if one of the two hypotheses is satisfied. The quantum state $\sigma_{i^\star}$ has the property to minimize the 1-norm between $\rho$ and $\{\sigma_i\}_i$, so it is natural to take the state minimizing the statistics of expected value roughly $\max_{i,j} \mathrm{tr}\, O_{i,j}(\rho - \sigma_l)$ for $l \in [m]$. To do this, we need to approximate $\mathrm{tr}\, \rho O_{i,j}$ for all $i \ne j$. We can use the classical shadow tomography algorithm of (Huang et al., 2020) to predict all these events using a few number of copies of $\rho$:

**Theorem 4.2.** *(Huang et al., 2020) Let $(O_1, \ldots, O_m)$ be a tuple of observables. There is an algorithm using non-adaptive incoherent measurements requiring:*

$$N = \mathcal{O}\left( \frac{d \log(m/\delta)}{\varepsilon^2} \right)$$

*copies of $\rho$ to predict $\mathrm{tr}(\rho O_i)$ to within $\varepsilon$-error for all $i = 1, \ldots, m$ with at most an error probability of $\delta$.*

Once we find the quantum state $\sigma_{i^\star}$, we know the basis of diagonalization of $\rho$. Hence we can learn the eigenvalues of the unknown quantum state $\rho$ by measuring it using the measurement device corresponding to its basis of diagonalization. This requires $\mathcal{O}(d/\varepsilon^2)$ incoherent copies. The algorithm is summarized in Alg. 1. This algorithm can be split in two phases. The first phase can be seen as an exploration phase, where the algorithm looks for the optimal eigen-basis. It collects (non-adaptively) the information given by the approximations $\mu_{i,j}$ of $\mathrm{tr}(\rho O_{i,j})$. Then it uses this information to choose $k^* = \mathrm{argmin}_l \max_{i,j} \mu_{i,j} - \mathrm{tr}(\sigma_l O_{i,j})$. After this step, in the second exploitation phase, the algorithm *adapts* its measurement device $\mathcal{M}$ according to the previous information $k^*$ and measures only with the POVM corresponding the the eigen-basis of $\sigma_{k^*}$.

Alg. 1 is $\delta$-correct (detailed proof deferred to App. B). It can be split in two parts for which we independently upper bound the copy complexity. The first part relies on the shadow tomography algorithm of (Huang et al., 2020) and needs a number

$$N_1 = \mathcal{O}\left( \frac{d \log(m(m-1)/\delta)}{(\varepsilon/10)^2} \right) = \mathcal{O}\left( \frac{d \log(m/\delta)}{\varepsilon^2} \right)$$

of copies of $\rho$. The second part requires a number $N_2 = 200 \frac{\log(2^{d+1}/\delta)}{\varepsilon^2}$ of copies of $\rho$. Finally, since $N = N_1 + N_2$ and $\delta = 1/3$, we have proven the following proposition:

**Proposition 4.3.** *Alg. 1 has a total copy complexity satisfying:*

$$N = \mathcal{O}\left( \frac{d \log(m)}{\varepsilon^2} \right).$$

### 4.1.2 Non-adaptive strategies.

We can slightly modify Alg. 1 to have a non-adaptive algorithm for the problem $(P)$ with incoherent measurements. It amounts to first measuring $\rho$ in all the basis corresponding to the known quantum states $(\sigma_i)_i$ and preparing $m$ approximated quantum states $(\tilde{\rho}_i)_i$. Then the tester can look for the closest quantum state $\sigma_{i^\star}$ and finally returns the approximated quantum state $\tilde{\rho}_{i^\star}$. This non-adaptive algorithm has a copy complexity

$$mN_2 + N_1 = \mathcal{O}\left(\frac{md + m\log(1/\delta) + d\log(m/\delta)}{\varepsilon^2}\right).$$

This complexity is almost optimal for $m \leq d$ (see Prop. 4.5). However, it is no longer optimal for $m \geq d$ since $md/\varepsilon^2 \geq d^2/\varepsilon^2$. In that case, we can still design an almost optimal non-adaptive algorithm as follows: for each $k \in [m]$, let $\{|\phi_i^k\rangle\}_i$ an orthonormal basis of diagonalization for $\sigma_k$. For each $k \in [m]$ and $B \subset [m]$, let $O_B^k = \sum_{i \in B} |\phi_i^k\rangle\langle\phi_i^k|$. We use the classical shadow tomography of (Huang et al., 2020) to predict $(\mathrm{tr}(\rho O_{i,j}))_{i,j \in [m]} \cup (\mathrm{tr}(\rho O_B^k))_{k \in [m], B \subset [m]}$ to within $\varepsilon/40$ simultaneously using

$$\mathcal{O}(d\log(m^2 + m2^d)/\varepsilon^2) = \mathcal{O}((d^2 + \log(m))/\varepsilon^2)$$

copies of $\rho$. We find the closest quantum state $\sigma_{i^\star}$ to $\rho$ the same way as the Alg. 1 does. Next, we look for a probability distribution $\tilde{\lambda}$ satisfying for all $B \subset [m] : \left|\tilde{\lambda}(B) - \mu_B^{i^\star}\right| \leq \varepsilon/40$, where $\mu_B^{i^\star}$ is the prediction of shadow tomography algorithm for $\mathrm{tr}(\rho O_B^{i^\star})$. Such $\tilde{\lambda}$ exists since the vector $\lambda$ of eigenvalues of $\rho$ satisfies the following property:

$$\mathrm{tr}(\rho O_B^{i^\star}) = \mathrm{tr}\left(\sum_{i \in [d]} \lambda_i |\phi_i^{i^\star}\rangle\langle\phi_i^{i^\star}| \sum_{i \in B} |\phi_i^{i^\star}\rangle\langle\phi_i^{i^\star}|\right) = \sum_{i \in [d], j \in B} \lambda_i |\langle\phi_i^{i^\star}|\phi_j^{i^\star}\rangle|^2 = \sum_{i \in B} \lambda_i = \lambda(B),$$

and $\left|\lambda(B) - \mu_B^{i^\star}\right| = \left|\mathrm{tr}(\rho O_B^{i^\star}) - \mu_B^{i^\star}\right| \leq \varepsilon/40$. We can thus return the quantum state $\tilde{\rho} = \sum_{i \in [d]} \tilde{\lambda}_i |\phi_i^{i^\star}\rangle\langle\phi_i^{i^\star}|$ as an approximation of $\rho$. We can verify that it is indeed an $\varepsilon/10$ approximation of $\rho$:

$$\|\rho - \tilde{\rho}\|_{\mathrm{tr}} \leq \sum_{i=1}^{d} |\lambda_i - \tilde{\lambda}_i| = 2\max_{B \subset [d]} \lambda(B) - \tilde{\lambda}(B)$$

$$\leq 2\max_{B \subset [d]} \lambda(B) - \mu_B^{i^\star} + 2\max_{B \subset [d]} \mu_B^{i^\star} - \tilde{\lambda}(B)$$

$$\leq 2\varepsilon/40 + 2\varepsilon/40 \leq \varepsilon/10.$$

The copy complexity of this algorithm is $\mathcal{O}((d^2 + \log(m))/\varepsilon^2$ which matches (up to logarithmic factors) the lower bound for $m \geq d$.

**Proposition 4.4.** *There is a non-adaptive algorithm for the the hypothesis selection problem $(P)$ with a total copy complexity satisfying:*

$$N = \mathcal{O}\left(\min\left\{\frac{d^2 + \log(m)}{\varepsilon^2}, \frac{md}{\varepsilon^2}\right\}\right).$$

## 4.2 Lower bound

In this section, we derive lower bounds for the problem $(P)$ both with adaptive and non-adaptive incoherent measurements. Note that the same lower bounds (up to constants) could be proven for sequential strategies as well. This is because if a sequential algorithm uses $N$ copies in expectation, then by Markov's inequality, it uses at most $10\,\mathbb{E}(N)$ copies with probability at least $9/10$.

We start with a lower bound for non-adaptive algorithms that matches the copy complexity of the algorithm presented in Sec. 4.1.2.

**Proposition 4.5.** *There is a tuple of quantum states $(\sigma_1, \ldots, \sigma_m)$ such that any learning algorithm with non-adaptive incoherent measurements requires*

$$N = \Omega\left(\min\left\{\frac{md}{\log(m)\varepsilon^2}, \frac{d^2}{\varepsilon^2}\right\}\right)$$

*copies of $\rho$ to approximate $\rho$ to at most $\varepsilon/10$ with at least a probability $2/3$.*

This result with $m = d$, together with the analysis of the adaptive Alg. 1 gives a nearly quadratic advantage for adaptive algorithms over non-adaptive ones.

*Sketch of the proof.* We construct a large set of quantum states randomly as follows: for $y \in \{1, \ldots, m\}$, $\sigma_y = U_y \Lambda U_y^\dagger = 2\frac{\mathbb{I}}{d} - \sigma_{m+1-y}$, where $U_y$ is a $d \times d$ unitary matrix $\mathrm{Haar}(d)$-distributed and $\Lambda$ is a diagonal matrix with entries $(1 \pm 10\varepsilon)/d$. Using the concentration inequality for Lipschitz functions of unitaries chosen according to the Haar measure, we can prove that this family is $\varepsilon$-separated with high probability:

**Lemma 4.6.** *Suppose that $m \leq \exp(d^2/3000)$. For $y \in [m/2]$, let $U_y \sim \mathrm{Haar}(d)$ and $\sigma_y = U_y \Lambda U_y^\dagger$. We have with at least a probability $9/10$, for all $y \neq z$:*

$$\|\sigma_y - \sigma_z\|_{\mathrm{tr}} > \varepsilon.$$

Then, for each $y$, we construct an $\varepsilon/10$-separated family of quantum states on the sphere of center $\sigma_y$ and radius $\varepsilon/3$ which have the same eigen-basis as $\sigma_y$. This can be done by taking random eigenvalues and using Hoeffding's inequality. This leads to a family of $e^{cd}$ states (for some constant $c$) that we denote by $\{\rho_{x,y}\}_{x \in [e^{cd}]}$.

By definition of the problem $(P)$, any $\delta$-correct non-adaptive algorithm for the problem $(P)$ can be used to distinguish between the states $\{\rho_{x,y}\}_{x,y}$ with probability at least $1 - \delta$. Thus, we can use these quantum states to encode a message in $[e^{cd}] \times [m]$ to a quantum state $\rho = \rho_{x,y}$ in the family constructed above. The decoder receives this unknown quantum state, performs non-adaptive incoherent measurements, and learns it. Therefore a $\delta$-correct algorithm can decode with a probability of failure at most $\delta$. By Fano's inequality, the encoder and decoder should share at least $\Omega(\log(m) + d)$ bits of information.

**Lemma 4.7** ((Fano, 1961)). *The mutual information between the encoder and the decoder is at least*

$$I \geq 2/3 \log(me^{cd}) - \log(2) \geq \Omega(\log(m) + d).$$

The remaining and crucial part of the proof is to upper bound the mutual information for a non-adaptive algorithm. After some manipulations, the use of Jensen's inequality, and some elementary inequalities of the logarithm function, we obtain an upper bound on the mutual information $I$ of the form:

$$I \leq \sup_{|\phi\rangle} \left( \frac{8N}{m \times e^{cd}} \sum_{y \in [m/2], x \in [e^{cd}]} \langle\phi|(d\rho_{x,y} - \mathbb{I})|\phi\rangle^2 \varepsilon^2 \right). \tag{2}$$

The next step is to show that the right hand side of the previous inequality cannot be bigger than $\mathcal{O}\left(\left(N\varepsilon^2 \left(\frac{\log(m)}{m} + \frac{1}{d}\right)\right)\right)$ with a probability at least $9/10$. This is the object of the following lemma.

**Lemma 4.8.** *By writing $\rho_{x,y} = \frac{\mathbb{I}}{d} + \frac{\varepsilon}{d} U_y O_{x,y} U_y^\dagger$ and $M = \frac{2}{m \times e^{cd}}$, we have with at least a probability $9/10$, for all unit vector $|\phi\rangle$:*

$$\frac{1}{M} \sum_{x \in [e^{cd}], y \in [m/2]} \langle\phi|(d\rho_{x,y} - \mathbb{I})|\phi\rangle^2 \leq \frac{C \log(m)}{m} + \frac{C}{d} + \frac{2\|O_{x,y}\|^2}{m}.$$

This lemma can be proven by considering a concentration inequality for the function

$$(U_y)_y \longmapsto \frac{2}{m \times e^{cd}} \sum_{x \in [e^{cd}], y \in [m/2]} \langle\phi|(d\rho_{x,y} - \mathbb{I})|\phi\rangle^2,$$

then considering a $1/m$-net on the unit sphere to deduce the required inequality for the previous function uniformly on the sphere. The proof uses techniques similar to the ones of (Haah et al., 2016) and (Chen et al., 2021). The detailed proof can be found in App. C.1.

A similar proof strategy allows to derive a lower bound on the copy complexity of adaptive strategies. The result is stated in the next proposition.

**Proposition 4.9.** *There is a tuple of quantum states* $(\sigma_1, \ldots, \sigma_m)$ *such that any learning algorithm with possibly adaptive incoherent measurements requires*

$$N = \Omega\left(\frac{d + \log(m)}{\varepsilon^2}\right)$$

*copies of $\rho$ to approximate $\rho$ to at most $\varepsilon/10$ with at least a probability $2/3$.*

The proof is similar to the one for non-adaptive strategies, with the minor difference that the adaptivity makes difficult to simplify some products, thus we cannot upper bound the mutual information as in Ineq. 2. We use instead a Cauchy Schwarz inequality to break the dependencies created by the adaptiveness of the algorithm. We obtain then an upper bound on the mutual information $I \leq \mathcal{O}(N\varepsilon^2)$. The detailed proof can be found in App. C.2.

This proposition along with the analysis of Alg. 1 show that the near optimal copy complexity of the problem $(P)$ using adaptive incoherent measurements is $\tilde{\Theta}\left(\frac{d}{\varepsilon^2}\right)$. This latter along with Prop. 4.5 imply the separation between adaptive and non-adaptive strategies for the problem $(P)$ for $m \gg 1$. In other words, knowing that the eigen-basis of the quantum state belongs to some family of bases gives an advantage to adaptive strategies since they can find the eigen-basis, and then focus on measuring the quantum state with the corresponding POVM. Up to our knowledge, this is the first example for which adaptive independent strategies outperform non-adaptive ones for testing quantum states.

## 5    Conclusion

We have constructed hypothesis selection problems for which sequential adaptive strategies are more efficient than non-sequential non-adaptive ones. The problem for which the advantage is the most significant is the one presented in Sec. 4. It would be interesting to see if there are other natural problems for which such a separation exists. We conjecture the separation would be polynomial in $m$ for the composite hypothesis selection problem: distinguishing between $\rho \in \{\sigma_1, \ldots, \sigma_m\}$ and $\rho \in \{\sigma_{m+1}, \ldots, \sigma_{2m}\}$ with high probability.

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

- Finally, we group the technical lemmas we often need in App. D.

## A  Reduction to classical testing problems

One of the main difficulties in quantum testing is the freedom in the choice of measurement at each step. So, to simplify the analysis of quantum testing problems, we provide some techniques permitting the reduction to classical testing problems which are well understood.

### A.1  Testing a single hypothesis vs a single hypothesis

#### A.1.1  Non-sequential strategies

We start by the simple remark that for diagonal quantum states, incoherent measurements can be seen as a post-processing of samples from the distribution given by the diagonal elements of the quantum state. Moreover, the stochastic map for this post-processing does not depend on the quantum state.

**Lemma A.1.** *Let $\mathcal{D}$ be a discrete distribution and $\rho$ its corresponding diagonal quantum state. Let $\mathcal{M}$ be a POVM. Measuring the quantum state $\rho$ with the POVM $\mathcal{M}$ can be seen as post-processing of samples from the distribution $\mathcal{D}$.*

*Proof.* Let $\mathcal{M} = \{M^i\}_{i \in [k]}$. For each $i \in [k]$, we can write

$$M^i = \sum_{x,y} M^i_{x,y} |x\rangle\langle y|.$$

By Born's rule, the probability distribution of the outcomes of the measurement of $\rho$ by the POVM $\mathcal{M}$ is:

$$\mathcal{M}(\rho) = \{\mathrm{tr}(\rho M^i)\}_{i \in d} = \left\{ \mathrm{tr}\left( \sum_x \mathcal{D}_x |x\rangle\langle x| \sum_{x,y} M^i_{x,y} |x\rangle\langle y| \right) \right\}_{i \in d}$$

$$= \left\{ \sum_x M^i_{x,x} \mathcal{D}_x \right\}_{i \in d} = \mathcal{P}\mathcal{D},$$

where $\mathcal{P} = (M^i_{x,x})_{i,x}$ is a stochastic matrix. Indeed, $M^i \succcurlyeq 0$ implies $M^i_{x,x} = \langle x|M^i|x\rangle \geq 0$ and $\sum_i M^i = \mathbb{I}$ implies

$$\sum_i M^i_{x,x} = \sum_i \langle x|M^i|x\rangle = \langle x|x\rangle = 1.$$

$\square$

Note that the post-processing map is independent of the quantum state, hence we can generalize the statement to any number of discrete distributions.

**Corollary A.2.** *Let $\mathcal{D}_1$ and $\mathcal{D}_2$ be two discrete distributions and $\rho_1$ and $\rho_2$ their corresponding diagonal quantum states. Let $\mathcal{M}$ be a POVM. Measuring the quantum state $\rho_1$ (resp. $\rho_2$) with the POVM $\mathcal{M}$ can be seen as post-processing (independent of the quantum states) of samples from the distribution $\mathcal{D}_1$ (resp. $\mathcal{D}_2$).*

We move now to the proof of the following upper and lower bound on discriminating two quantum states using non sequential incoherent measurements:

**Proposition A.3.** *There is a non-sequential algorithm for testing $H_1 : \rho = \sigma_1$ vs $H_2 : \rho = \sigma_2$ using a number of measurements*

$$N \leq \frac{2\log(1/\delta)}{\varepsilon^2}.$$

*Moreover, there exists two quantum states $\sigma_1$ and $\sigma_2$ satisfying $\|\sigma_1 - \sigma_2\|_{\mathrm{tr}} = \varepsilon$ so that every non-sequential algorithm distinguishing between $H_1 : \rho = \sigma_1$ and $H_2 : \rho = \sigma_2$ needs a number of measurements satisfying*

$$\liminf_{\delta \to 0} \frac{N}{\log(1/\delta)} \geq \max\left\{ \frac{1}{\mathrm{KL}(1/2 + \alpha\varepsilon\|1/2)}, \frac{1}{\mathrm{KL}(1/2 - \beta\varepsilon\|1/2)} \right\} \underset{\varepsilon \to 0}{\sim} \frac{2}{\varepsilon^2},$$

*where $\alpha \in (0,1)$ and $\beta \in (0,1)$ are defined such that $\mathrm{KL}(1/2 + \alpha\varepsilon\|1/2) = \mathrm{KL}(1/2 + \alpha\varepsilon\|1/2 + \varepsilon)$ and $\mathrm{KL}(1/2 - \beta\varepsilon\|1/2) = \mathrm{KL}(1/2 - \beta\varepsilon\|1/2 - \varepsilon)$.*

*Proof.* The correctness of the batch algorithm presented in Sec. 3.1.1 can be done using Chernoff-Hoeffding inequality, if $\rho = \sigma_1$ the error probability can be upper bounded as follows:

$$
\begin{aligned}
\mathbb{P}\left(S - \mathrm{tr}(\sigma_2 O) \leq \varepsilon/2\right) = \mathbb{P}\left(S - \mathrm{tr}(\sigma_1 O) \leq \varepsilon/2 - \varepsilon\right) \\
\leq \mathbb{P}\left(S - \mathrm{tr}(\sigma_1 O) \leq -\varepsilon/2\right) \\
\leq \exp(-N\,\mathrm{KL}(\mathrm{tr}(\sigma_1 O) - \varepsilon/2\|\mathrm{tr}(\sigma_1 O))).
\end{aligned}
$$

On the other hand, if $\rho = \sigma_2$:

$$\mathbb{P}\left(S - \mathrm{tr}(\sigma_2 O) \geq \varepsilon/2\right) \leq \exp(-N\,\mathrm{KL}(\mathrm{tr}(\sigma_2 O) + \varepsilon/2\|\mathrm{tr}(\sigma_2 O))).$$

Therefore to ensure that the batch algorithm is $\delta$-correct we need $N$ to satisfy

$$N = \max\left\{ \frac{\log(1/\delta)}{\mathrm{KL}(\mathrm{tr}(\sigma_1 O) - \varepsilon/2\|\mathrm{tr}(\sigma_1 O))}, \frac{\log(1/\delta)}{\mathrm{KL}(\mathrm{tr}(\sigma_2 O) + \varepsilon/2\|\mathrm{tr}(\sigma_2 O))} \right\}.$$

Moreover by Pinsker's inequality (Fedotov et al., 2003), the right hand side is upper bounded by:

$$\max\left\{ \frac{\log(1/\delta)}{\mathrm{KL}(\mathrm{tr}(\sigma_1 O) - \varepsilon/2\|\mathrm{tr}(\sigma_1 O))}, \frac{\log(1/\delta)}{\mathrm{KL}(\mathrm{tr}(\sigma_2 O) + \varepsilon/2\|\mathrm{tr}(\sigma_2 O))} \right\} \leq \frac{2\log(1/\delta)}{\varepsilon^2}.$$

For the lower bound, let $d = 2$, $\sigma_1 = \mathbb{I}/2$ and $\sigma_2 = \mathrm{diag}((1 + 2\varepsilon)/2, (1 - 2\varepsilon)/2) = \mathbb{I}/2 + \varepsilon O$ where $O = \mathrm{diag}(1, -1)$. Let $\mathcal{A}$ be a non sequential algorithm that distinguishes between $H_1$ and $H_2$ using $N$ measurements. Let the $i^{\mathrm{th}}$ measurement be $\mathcal{M}_i = (\mathbb{I} - O_i, O_i)$. Measuring $\rho = \sigma_1$ (resp. $\sigma_2$) with the POVM $\mathcal{M}_i = (\mathbb{I} - O_i, O_i)$ is equivalent to sampling from $\mathrm{Ber}(\mathrm{tr}(O_i)/2)$ (resp. $\mathrm{Ber}(\mathrm{tr}(O_i)/2 + \varepsilon\mathrm{tr}(O_i O))$). The optimal sample complexity of testing identity: $H_0 : p = \mathrm{tr}(O_i)/2$ vs $H_1 : p = \mathrm{tr}(O_i)/2 + \varepsilon\mathrm{tr}(O_i O)$ is asymptotically equivalent to (when $\varepsilon \to 0$) (Fawzi et al., 2022):

$$\frac{8(\mathrm{tr}(O_i)/2)(1 - \mathrm{tr}(O_i)/2)\log(1/\delta)}{\varepsilon^2 \mathrm{tr}(O_i O)^2} = \frac{4\mathrm{tr}(O_i)(1 - \mathrm{tr}(O_i)/2)\log(1/\delta)}{\varepsilon^2 \mathrm{tr}(O_i O)^2}.$$

Let's write $O_i = \begin{pmatrix} \lambda_1 & \beta \\ \bar{\beta} & \lambda_2 \end{pmatrix}$, we have $\mathrm{tr}(O_i O) = \lambda_1 - \lambda_2$. Since $0 \preccurlyeq O_i \preccurlyeq \mathbb{I}$, we have $0 \leq \lambda_i \leq 1$ for $i = 1, 2$. Hence

$$\frac{4\mathrm{tr}(O_i)(1 - \mathrm{tr}(O_i)/2)\log(1/\delta)}{\varepsilon^2 \mathrm{tr}(O_i O)^2} = \frac{2(\lambda_1 + \lambda_2)(2 - \lambda_1 - \lambda_2)\log(1/\delta)}{\varepsilon^2(\lambda_1 - \lambda_2)^2} \geq \frac{2\log(1/\delta)}{\varepsilon^2}.$$

This latter inequality is true since

$$(\lambda_1 + \lambda_2)(2 - \lambda_1 - \lambda_2) \geq (\lambda_1 - \lambda_2)^2 \iff \lambda_1(1 - \lambda_1) + \lambda_2(1 - \lambda_2) \geq 0,$$

with equality iff $\lambda_i = 0, 1$ for $i = 1, 2$. The cases $\lambda_1 = \lambda_2$ are eliminated because the sample complexity has a denominator $(\lambda_1 - \lambda_2)$. It remains the cases $\lambda_1 = 1 - \lambda_2 \in \{0, 1\}$ for which $O_i$ is a rank 1 projector. Therefore, the optimal measurement reduces to testing uniform: $\mathrm{Ber}(1/2)$ vs $\mathrm{Ber}(1/2 \pm \varepsilon)$. This problem requires a sample complexity asymptotically equivalent to $\frac{2 \log(1/\delta)}{\varepsilon^2}$. Note that we can also use Lemma A.1 to make the desired reduction. We show how this reduction works for entangled strategies. We have $\sigma_1^{\otimes N} = \frac{\mathbb{I}}{2^N}$ and $\sigma_2^{\otimes N} = \frac{1}{2^N} \mathrm{diag}\left((1 + 2\varepsilon)^{|i|}(1 - 2\varepsilon)^{N-|i|}\right)_{i \in \{0,1\}^N}$ where $|i| = i_1 + \cdots + i_N$. By Lemma A.1, measuring the quantum states $\sigma_1^{\otimes N}$ (resp. $\sigma_2^{\otimes N}$) can be seen as post-processing of samples from the distribution $\mathcal{D}_1 = \{1/2^N\}_{i \in \{0,1\}^N}$ (resp. $\mathcal{D}_2 = \left\{(1/2 - \varepsilon)^{|i|}(1/2 + \varepsilon)^{N-|i|}\right\}_{i \in \{0,1\}^N}$). Observe that a sample $i = (i_1, \ldots, i_N) \sim \mathcal{D}_1$ is given by $N$ i.i.d. random variables $\{i_k \sim \mathrm{Ber}(1/2)\}_{k \in [N]}$. Similarly, a sample $i = (i_1, \ldots, i_N) \sim \mathcal{D}_2$ is given by $N$ i.i.d. random variables $\{i_k \sim \mathrm{Ber}(1/2 - \varepsilon)\}_{k \in [N]}$. Therefore, distinguishing $\sigma_1$ from $\sigma_2$ using $N$ entangled copies can be reduced to testing $\mathrm{Ber}(1/2)$ vs $\mathrm{Ber}(1/2 - \varepsilon)$ using $N$ samples. This latter requires a number of samples (Fawzi et al., 2022):

$$\liminf_{\delta \to 0} \frac{N}{\log(1/\delta)} \geq \max\left\{\frac{1}{\mathrm{KL}(1/2 + \alpha\varepsilon \| 1/2)}, \frac{1}{\mathrm{KL}(1/2 - \beta\varepsilon \| 1/2)}\right\} \underset{\varepsilon \to 0}{\sim} \frac{2}{\varepsilon^2},$$

where $\alpha \in (0, 1)$ and $\beta \in (0, 1)$ are defined such that $\mathrm{KL}(1/2 + \alpha\varepsilon \| 1/2) = \mathrm{KL}(1/2 + \alpha\varepsilon \| 1/2 + \varepsilon)$ and $\mathrm{KL}(1/2 - \beta\varepsilon \| 1/2) = \mathrm{KL}(1/2 - \beta\varepsilon \| 1/2 - \varepsilon)$.

$\square$

### A.1.2 Sequential strategies

Discriminating two quantum states using sequential strategies can be done with fewer measurements than non-sequential strategies. Since the reduction to lower bound is similar, we give only the proof for the upper bound.

**Proposition A.4.** *There is a sequential algorithm for testing $H_1 : \rho = \sigma_1$ vs $H_2 : \rho = \sigma_2$ using an expected number of measurements:*

$$\mathbb{E}(N) \leq \frac{\log(1/\delta)}{2\varepsilon^2} + \frac{\log(1/\delta)^{2/3} + 2\log(1/\delta)^{1/3} + \log(\log(1/\delta)/2\varepsilon^2) + 1}{\varepsilon^2}.$$

*Moreover, there are two quantum states $\sigma_1$ and $\sigma_2$ satisfying $\|\sigma_1 - \sigma_2\| = \varepsilon$ so that every sequential algorithm distinguishing between $H_1 : \rho = \sigma_1$ and $H_2 : \rho = \sigma_2$ with high probability needs in expectation a number*

$$\mathbb{E}(N) \geq \frac{\log(1/\delta)}{\min\{\mathrm{KL}(1/2 \pm \varepsilon \| 1/2)\}}$$

*of measurements.*

*Proof.* The algorithm is presented in Sec. 3.1.2.

**Correctness.** Let's start by showing that this algorithm is $\delta$-correct. To this end, we need a time uniform concentration inequality which can be obtained by Hoeffding inequality along with the union bound, recall that $S_t = (\sum_{i=1}^t X_i)/t$ and $X_i \sim \mathrm{Ber}(\mathrm{tr}(\rho O))$:

$$\mathbb{P}\left(\exists t \geq 1 : |S_t - \mathbb{E}(S_t)| > \phi(\delta, t)\right) \leq \sum_{t \geq 1} \mathbb{P}\left(|S_t - \mathbb{E}(S_t)| > \phi(\delta, t)\right)$$
$$\leq \sum_{t \geq 1} \exp(-2t\phi(\delta, t)^2)$$
$$\leq \sum_{t \geq 1} \frac{\delta}{t(t+1)}$$
$$\leq \delta.$$

**Complexity.** To obtain an upper bound on the complexity, we use the following lemma:

**Lemma A.5.** $N$ *a random variable taking values in* $\mathbb{N}$*, we have for all* $k \in \mathbb{N}^*$

$$\mathbb{E}(N) \leq k + \sum_{t \geq k} \mathbb{P}(N \geq t) \,.$$

This inequality can be proved by writing $\mathbb{E}(N) = \sum_{t \geq 0} \mathbb{P}(N \geq t)$ then upper bounding the first $k$ terms by 1.

Let $\alpha \in (0,1)$ and $k$ the smallest integer so that for all $t \geq k : \phi(\delta, t) \leq \alpha\varepsilon$. We focus only on the case $\rho = \sigma_1$ (the other being similar), the expected stopping time of the algorithm can be controlled as follows:

$$
\begin{aligned}
\mathbb{E}(N) &\leq k + \sum_{t \geq k} \mathbb{P}(N \geq t) \\
&\leq k + \sum_{t \geq k} \mathbb{P}(S_{t-1} < \operatorname{tr}(\sigma_2 O) + \phi(\delta, t-1)) \\
&\leq k + \sum_{t \geq k-1} \mathbb{P}(S_t - \operatorname{tr}(\sigma_1 O) < -\varepsilon + \alpha\varepsilon) \\
&\leq k + \sum_{t \geq k-1} \mathbb{P}(S_t - \operatorname{tr}(\sigma_1 O) < -(1-\alpha)\varepsilon) \\
&\leq k + \sum_{t \geq k-1} 2 \exp(-2t(1-\alpha)^2\varepsilon^2) \\
&\leq k + \frac{2 \exp(-2(k-1)(1-\alpha)^2\varepsilon^2)}{1 - \exp(-2(1-\alpha)^2\varepsilon^2)} \\
&\leq k + \frac{2 \exp(-2(k-1)(1-\alpha)^2\varepsilon^2)}{(1-\alpha)^2\varepsilon^2} \,.
\end{aligned}
$$

On the other hand we have $\phi(\delta, k) \leq \alpha\varepsilon$ and $\phi(\delta, k-1) \geq \alpha\varepsilon$ so

$$\log\left(\frac{(k-1)k}{\delta}\right) \geq 2(k-1)\alpha^2\varepsilon^2.$$

Therefore:

$$k - 1 \leq \frac{\log(1/\delta)}{2\alpha^2\varepsilon^2} + 2\frac{\log(\log(1/\delta)/(\alpha\varepsilon)^2)}{\alpha^2\varepsilon^2}.$$

Hence:

$$\frac{\mathbb{E}(N)}{\log(1/\delta)} \leq \frac{1}{2\alpha^2\varepsilon^2} + 2\frac{\log(\log(1/\delta)/(\alpha\varepsilon)^2)}{\log(1/\delta)\alpha^2\varepsilon^2} + \frac{1}{\log(1/\delta)} + \frac{2 \exp(-2(k-1)(1-\alpha)^2\varepsilon^2)}{\log(1/\delta)(1-\alpha)^2\varepsilon^2},$$

and by taking $\delta \to 0$, then $\alpha \to 1$ we obtain:

$$\limsup_{\delta \to 0} \frac{\mathbb{E}(N)}{\log(1/\delta)} \leq \frac{1}{2\varepsilon^2}.$$

A non asymptotic upper bound can be obtained by choosing $\alpha = (1 + \log(1/\delta)^{-1/3})^{-1}$:

$$\mathbb{E}(N) \leq \frac{\log(1/\delta)}{2\varepsilon^2} + \frac{\log(1/\delta)^{2/3} + 2\log(1/\delta)^{1/3} + \log(\log(1/\delta)/2\varepsilon^2) + 1}{\varepsilon^2}.$$

The lower bound follows from the previous reduction to testing $\mathrm{Ber}(1/2)$ vs $\mathrm{Ber}(1/2 \pm \varepsilon)$ and (Fawzi et al., 2022). $\qquad\square$

### A.1.3 Asymptotics of the quantum relative entropy and Chernoff divergence.

Recall that we consider the particular states $\sigma_1 = \frac{\mathbb{I}_2}{2}$ and $\sigma_2 = \mathrm{diag}(\frac{1}{2} + \varepsilon, \frac{1}{2} - \varepsilon)$. An asymptotic (when $\varepsilon \to 0$) of the quantum relative entropy between $\sigma_1$ and $\sigma_2$ is given by:

$$D(\sigma_1 \| \sigma_2) = \frac{1}{2} \log\left(\frac{1}{1 + 2\varepsilon}\right) + \frac{1}{2} \log\left(\frac{1}{1 - 2\varepsilon}\right) \underset{\varepsilon \to 0}{\sim} 2\varepsilon^2,$$

$$D(\sigma_2 \| \sigma_1) = \left(\frac{1}{2} + \varepsilon\right) \log(1 + 2\varepsilon) + \left(\frac{1}{2} - \varepsilon\right) \log(1 - 2\varepsilon) \underset{\varepsilon \to 0}{\sim} 2\varepsilon^2.$$

On the other hand, an asymptotic (when $\varepsilon \to 0$) of the quantum Chernoff divergence between the states $\sigma_1$ and $\sigma_2$ can be upper bounded using the inequality $\log(x) \geq (x - 1) - (x - 1)^2$ valid for $x \in (\frac{1}{2}, \infty)$:

$$\begin{aligned}
C(\sigma_1, \sigma_2) &= \sup_{0 \leq s \leq 1} -\log(\mathrm{tr}(\sigma_1^s \sigma_2^{1-s})) \\
&= \sup_{0 \leq s \leq 1} -\log\left(\frac{1}{2}(1 + 2\varepsilon)^s + \frac{1}{2}(1 - 2\varepsilon)^s\right) \\
&\leq \sup_{0 \leq s \leq 1} \left[1 - \frac{1}{2}(1 + 2\varepsilon)^s - \frac{1}{2}(1 - 2\varepsilon)^s\right] + \left[\frac{1}{2}(1 + 2\varepsilon)^s + \frac{1}{2}(1 - 2\varepsilon)^s - 1\right]^2 \\
&\leq \sup_{0 \leq s \leq 1} \left[2s(1 - s)\varepsilon^2 + o(\varepsilon^2)\right] + o(\varepsilon^4) \underset{\varepsilon \to 0}{\sim} \frac{\varepsilon^2}{2}.
\end{aligned}$$

Moreover, it can be lower bounded using the inequality $\log(x) \leq x - 1$:

$$\begin{aligned}
C(\sigma_1, \sigma_2) &= \sup_{0 \leq s \leq 1} -\log\left(\frac{1}{2}(1 + 2\varepsilon)^s + \frac{1}{2}(1 - 2\varepsilon)^s\right) \\
&\geq \sup_{0 \leq s \leq 1} \left[1 - \frac{1}{2}(1 + 2\varepsilon)^s - \frac{1}{2}(1 - 2\varepsilon)^s\right] \\
&\geq \sup_{0 \leq s \leq 1} \left[2s(1 - s)\varepsilon^2 + o(\varepsilon^2)\right] \underset{\varepsilon \to 0}{\sim} \frac{\varepsilon^2}{2}.
\end{aligned}$$

Finally $C(\sigma_1, \sigma_2) \underset{\varepsilon \to 0}{\sim} \frac{\varepsilon^2}{2}$.

### A.2 Testing a single hypothesis vs a multiple hypothesis

In this section, we relate the TV-distance between the distributions obtained after the measurements and the 1-norm between the quantum states.

**Lemma A.6.** *Let $U \in \mathbb{C}^{d \times d}$ be a* Haar-*random unitary matrix of columns $\{|U_i\rangle\}_{1 \leq i \leq d}$, $\mathcal{M}(U) = \{|U_i\rangle\langle U_i|\}_i$ is a POVM and there exists a universal constant $c$ such that for all quantum states $\rho$ and $\sigma$ we have:*

$$\mathbb{E}\left[\mathrm{TV}(\rho(\mathcal{M}), \sigma(\mathcal{M}))\right] \geq c \frac{\|\rho - \sigma\|_{\mathrm{tr}}}{\sqrt{d}}.$$

*Proof.* Let $\xi = \rho - \sigma$, we have $U|e_i\rangle = |U_i\rangle$ and we use Weingarten Calculus D.1 and D.2 to calculate

$$\begin{aligned}
\mathbb{E}\left[\langle U_i|\xi|U_i\rangle^2\right] &= \mathbb{E}\left[\langle U_i|\xi|U_i\rangle\langle U_i|\xi|U_i\rangle\right] \\
&= \mathbb{E}\left[\mathrm{tr}(\xi|U_i\rangle\langle U_i|\xi|U_i\rangle\langle U_i|)\right] \\
&= \mathbb{E}\left[\mathrm{tr}(\xi U|e_i\rangle\langle e_i|U^*\xi U|e_i\rangle\langle e_i|U^*)\right] \\
&= \mathbb{E}\left[\mathrm{tr}(U^*\xi U|e_i\rangle\langle e_i|U^*\xi U|e_i\rangle\langle e_i|)\right] \\
&= \sum_{\alpha, \beta \in \mathfrak{S}_2} \mathrm{Wg}(\beta\alpha^{-1}, d)\mathrm{tr}_{\beta^{-1}}(\xi, \xi)\mathrm{tr}_\alpha(|e_i\rangle\langle e_i|, |e_i\rangle\langle e_i|) \\
&= \frac{1}{d(d + 1)}\mathrm{tr}(\xi^2).
\end{aligned}$$

Similarly

$$
\begin{aligned}
\mathbb{E}\left[\langle U_i|\xi|U_i\rangle^4\right] &= \mathbb{E}\left[\langle U_i|\xi|U_i\rangle\langle U_i|\xi|U_i\rangle\langle U_i|\xi|U_i\rangle\langle U_i|\xi|U_i\rangle\right] \\
&= \mathbb{E}\left[\mathrm{tr}(\xi|U_i\rangle\langle U_i|\xi|U_i\rangle\langle U_i|\xi|U_i\rangle\langle U_i|\xi|U_i\rangle\langle U_i|)\right] \\
&= \mathbb{E}\left[\mathrm{tr}(\xi U|e_i\rangle\langle e_i|U^*\xi U|e_i\rangle\langle e_i|U^*\xi U|e_i\rangle\langle e_i|U^*\xi U|e_i\rangle\langle e_i|U^*)\right] \\
&= \mathbb{E}\left[\mathrm{tr}(U^*\xi U|e_i\rangle\langle e_i|U^*\xi U|e_i\rangle\langle e_i|U^*\xi U|e_i\rangle\langle e_i|U^*\xi U|e_i\rangle\langle e_i|)\right] \\
&= \sum_{\alpha,\beta\in\mathfrak{S}_4} \mathrm{Wg}(\beta\alpha^{-1},d)\mathrm{tr}_{\beta^{-1}}(\xi,\xi,\xi,\xi)\mathrm{tr}_\alpha(|e_i\rangle\langle e_i|,|e_i\rangle\langle e_i|,|e_i\rangle\langle e_i|,|e_i\rangle\langle e_i|) \\
&= \frac{1}{d(d+1)(d+2)(d+3)}(6\mathrm{tr}(\xi^2)^2 + 6\mathrm{tr}(\xi^4)). \\
&\leq \frac{c'}{d(d+1)(d+2)(d+3)}\mathrm{tr}(\xi^2)^2.
\end{aligned}
$$

We can now conclude by Hölder's inequality:

$$
\begin{aligned}
2\mathbb{E}\left[\mathrm{TV}(\rho(\mathcal{M}),\sigma(\mathcal{M}))\right] &= \sum_{i=1}^{d}\mathbb{E}\left[|\langle U_i|\xi|U_i\rangle|\right] \\
&\geq \sum_{i=1}^{d}\sqrt{\frac{\left(\mathbb{E}\left[\langle U_i|\xi|U_i\rangle^2\right]\right)^3}{\mathbb{E}\left[\langle U_i|\xi|U_i\rangle^4\right]}} \\
&\geq \sum_{i=1}^{d}\sqrt{\frac{(d^{-1}(d+1)^{-1}\mathrm{tr}(\xi^2))^3}{c'd^{-1}(d+1)^{-1}(d+2)^{-1}(d+3)^{-1}\mathrm{tr}(\xi^2)^2}} \\
&\geq \sum_{i=1}^{d}c\frac{\sqrt{\mathrm{tr}(\xi^2)}}{d} \\
&\geq c\sqrt{\mathrm{tr}(\rho-\sigma)^2}.
\end{aligned}
$$

$\square$

This Lemma is about the expected TV distance. Actually, we can prove that we have the same inequality with high probability.

**Lemma A.7.** *Let $l = \Theta\left(\log(1/\delta)\right)$ and $U^1, U^2, \ldots, U^l \in \mathbb{C}^{d\times d}$ be* Haar-*random unitary matrices of columns $\{|U_i^j\rangle\}_{1\leq i\leq d, 1\leq j\leq l}$, $\mathcal{M} = \{\frac{1}{l}|U_i^j\rangle\langle U_i^j|\}_{i,j}$ is a POVM and there exists a universal constant $c$ such that for all quantum states $\rho$ and $\sigma$ we have with a probability at least $1-\delta$:*

$$
\mathrm{TV}(\rho(\mathcal{M}),\sigma(\mathcal{M})) \geq c\frac{\|\rho-\sigma\|_{\mathrm{tr}}}{\sqrt{d}}.
$$

*Proof.* Let $f(U) = \mathrm{TV}(\rho(\mathcal{M}), \sigma(\mathcal{M}))$, we first show that $f$ is Lipschitz by using the triangle and Cauchy Schwarz inequalities:

$$
\begin{aligned}
2|f(U) - f(V)| &= \left| \sum_{1 \le i \le d, 1 \le j \le l} \frac{1}{l} |\mathrm{tr}(|U_i^j\rangle\langle U_i^j|\xi)| - |\mathrm{tr}(|V_i^j\rangle\langle V_i^j|\xi)| \right| \\
&\le \sum_{1 \le i \le d, 1 \le j \le l} \frac{1}{l} \left| \mathrm{tr}((|U_i^j\rangle\langle U_i^j| - |V_i^j\rangle\langle V_i^j|)\xi) \right| \\
&\le \sum_{1 \le i \le d, 1 \le j \le l} \frac{1}{l} \sqrt{\mathrm{tr}(\xi^2)} \sqrt{\mathrm{tr}((|U_i^j\rangle\langle U_i^j| - |V_i^j\rangle\langle V_i^j|)^2)} \\
&\le \sqrt{\frac{d}{l}} \sqrt{\mathrm{tr}(\xi^2)} \sqrt{\sum_{1 \le i \le d, 1 \le j \le l} \mathrm{tr}((|U_i^j\rangle\langle U_i^j| - |V_i^j\rangle\langle V_i^j|)^2)} \\
&\le \sqrt{\frac{d}{l}} \sqrt{\mathrm{tr}(\xi^2)} \sqrt{\sum_{1 \le j \le l} \mathrm{tr}((U^j - V^j)^2)} \\
&\le \sqrt{\frac{d}{l}} \sqrt{\mathrm{tr}(\xi^2)} \|U - V\|_{2,\mathrm{HS}},
\end{aligned}
$$

hence $f$ is $L = \sqrt{\frac{d}{2l}} \sqrt{\mathrm{tr}(\xi^2)}$-Lipschitz, therefore by Theorem D.3:

$$
\mathbb{P}\left( |f(U) - \mathbb{E}(f(U))| > \frac{c}{2} \sqrt{\mathrm{tr}(\xi^2)} \right) \le e^{-\frac{dc^2 \mathrm{tr}(\xi^2)}{48 L^2}} = e^{-lc^2/24} = \delta/2,
$$

for $l = 24 \log(2/\delta)/c^2$. Finally with high probability (at least $1 - \delta/2$) we have

$$
\begin{aligned}
\mathrm{TV}(\rho(\mathcal{M}), \sigma(\mathcal{M})) &\ge \mathbb{E}(\mathrm{TV}(\rho(\mathcal{M}), \sigma(\mathcal{M}))) - |\mathrm{TV}(\rho(\mathcal{M}), \sigma(\mathcal{M})) - \mathbb{E}(\mathrm{TV}(\rho(\mathcal{M}), \sigma(\mathcal{M})))| \\
&\ge c\sqrt{\mathrm{tr}(\xi^2)} - \frac{c}{2}\sqrt{\mathrm{tr}(\xi^2)} \\
&\ge \frac{c}{2}\sqrt{\mathrm{tr}(\xi^2)} \\
&\ge \frac{c}{2} \frac{\|\rho - \sigma\|_{\mathrm{tr}}}{\sqrt{r}},
\end{aligned}
$$

where $r$ is the rank of $(\rho - \mathbb{I}/d)$. $\qquad\square$

Once we have the lower bound on the TV distance between the distributions obtained after performing the measurements, we can deduce upper bounds on sequential algorithms for testing identity depending on the rank of $\rho$ or $\rho - \mathbb{I}/d$.

**Dependence in the rank of $\rho - \mathbb{I}/d$**  From the previous lower bound on the TV-distance, we can achieve an upper bound using the sequential tester of (Fawzi et al., 2022):

$$
\begin{aligned}
&\mathcal{O}\left( \min\left\{ \frac{n^{1/2} \log(1/\delta)^{1/2}}{(\max\{\varepsilon/\sqrt{d}, \|\rho - \mathbb{I}/d\|_2\})^2}, \frac{\log(1/\delta)}{(\max\{\varepsilon/\sqrt{d}, \|\rho - \mathbb{I}/d\|_2\})^2} \right\} \right) = \mathcal{O}\left( \frac{d^{3/2} \log(1/\delta)}{\max\{\varepsilon^2, d\|\rho - \mathbb{I}/d\|_2^2\}} \right) \\
&= \mathcal{O}\left( \min\left\{ \frac{d^{3/2} \log(1/\delta)}{\varepsilon^2}, \frac{rd^{1/2} \log(1/\delta)}{\|\rho - \mathbb{I}/d\|_{\mathrm{tr}}^2} \right\} \right),
\end{aligned}
$$

where $r$ is the rank of $(\rho - \mathbb{I}/d)$ and we use Cauchy Schwarz to obtain the latter inequality.

**Dependence in the rank of $\rho$**  The proof of Lemma 3.5 permits to deduce that with high probability:

$$\mathrm{TV}(P, U_n) \geq c\|\rho - \mathbb{I}/d\|_2 \geq c\sqrt{\sum_{i=1}^{r}\left(\lambda_i - \frac{1}{d}\right)^2 + \frac{d-r}{d^2}}$$

$$\geq c\sqrt{\sum_{i=1}^{r}\lambda_i^2 - \frac{1}{d}} \geq c\sqrt{\frac{1}{r} - \frac{1}{d}} \geq c\sqrt{\frac{1}{2r}},$$

where $r$ is the rank of $\rho$ supposed to be less than $d/2$ and we use Cauchy Schwarz inequality. Therefore we can test whether $\rho = \sigma$ or $\|\rho - \sigma\|_{\mathrm{tr}} > \varepsilon$ with probability at least $1 - \delta$ using

$$\mathcal{O}\left(\frac{d^{1/2}\log(1/\delta)}{\left(\max\left\{\varepsilon/\sqrt{d}, 1/\sqrt{2r}\right\}\right)^2}\right) = \mathcal{O}\left(\min\left\{\frac{d^{3/2}\log(1/\delta)}{\varepsilon^2}, rd^{1/2}\log(1/\delta)\right\}\right)$$

copies of $\rho$.

# B  Analysis of Alg. 1

In this section we prove the correctness of the Alg. 1. We need to show that with probability at least $1 - \delta/2$, Alg. 1 finds the closest quantum state $\sigma_{i^*}$ to $\rho$.

**Lemma B.1.** *For all $i \neq j \in [m]$, let $\mu_{i,j}$ an $\varepsilon/10$ approximation of $\mathrm{tr}(\rho O_{i,j})$ given by classical shadow tomography of (Huang et al., 2020). Let $k^* = \mathrm{argmin}_l \max_{i,j} \mu_{i,j} - \mathrm{tr}(\sigma_l O_{i,j})$. We have with at least a probability $1 - \delta/2$:*

$$\|\rho - \sigma_{k^*}\|_{\mathrm{tr}} \leq \varepsilon/3.$$

*Proof.* Classical shadow tomography of (Huang et al., 2020) permits to have the following approximations

$$\forall i \neq j \in [m] : |\mu_{i,j} - \mathrm{tr}(\rho O_{i,j})| \leq \varepsilon/10,$$

with a probability at least $1 - \delta/2$ using only $N = \mathcal{O}(d\log(m)/\varepsilon^2)$ copies of $\rho$.

Let $\sigma_{i^*}$ the closest quantum state to $\rho$. We want to prove that with high probability $k^* = i^*$. We have for all $l \neq i^*$: $\|\sigma_{i^*} - \sigma_l\|_{\mathrm{tr}} > \varepsilon$ hence:

$$\max_{i,j}\mu_{i,j} - \mathrm{tr}(\sigma_l O_{i,j}) \geq \mu_{i^*,l} - \mathrm{tr}(\sigma_l O_{i^*,l})$$

$$\geq \mathrm{tr}(\rho O_{i^*,l}) - \mathrm{tr}(\sigma_l O_{i^*,l}) - \varepsilon/10$$
$$\geq \mathrm{tr}(\sigma_{i^*} O_{i^*,l}) - \mathrm{tr}(\sigma_l O_{i^*,l}) + \mathrm{tr}(\rho O_{i^*,l}) - \mathrm{tr}(\sigma_{i^*} O_{i^*,l}) - \varepsilon/10$$
$$\geq \|\sigma_{i^*} - \sigma_l\|_{\mathrm{tr}} - \|\rho - \sigma_{i^*}\|_{\mathrm{tr}} - \varepsilon/10$$
$$\geq \varepsilon - \varepsilon/3 - \varepsilon/10$$
$$> \varepsilon/2.$$

On the other hand

$$\max_{i,j}\mu_{i,j} - \mathrm{tr}(\sigma_{k^*} O_{i,j}) \leq \max_{i,j}\mu_{i,j} - \mathrm{tr}(\sigma_{i^*} O_{i,j})$$

$$\leq \max_{i,j}\mathrm{tr}(\rho O_{i,j}) - \mathrm{tr}(\sigma_{i^*} O_{i,j}) + \varepsilon/10$$
$$\leq \|\rho - \sigma_{i^*}\|_{\mathrm{tr}} + \varepsilon/10$$
$$\leq \varepsilon/3 + \varepsilon/10$$
$$< \varepsilon/2.$$

Therefore, with high probability, $k^*$ cannot be different from $i^*$.  $\square$

Once we know, with high probability, the closest quantum state to $\rho$ we can read its basis and use it to to learn $\rho$. The following lemma indicates how to construct this approximation along with the number of copies/measurements needed for this learning task.

**Lemma B.2.** *Let $\rho = \sum_{i=1}^{d} \lambda_i |\phi_i\rangle\langle\phi_i|$. Let $A_1, \ldots, A_N$ the outcomes of the measurement of $\rho$ independently by the POVM $\mathcal{M} = \{|\phi_i\rangle\langle\phi_i|\}_i$. The quantum state*

$$\tilde{\rho} = \sum_{i=1}^{d} \left( \frac{\sum_{j=1}^{N} \mathbf{1}_{A_j = i}}{N} \right) |\phi_i\rangle\langle\phi_i|$$

*is $\varepsilon/10$-close in 1-norm to $\rho$ with a probability at least $1 - \delta/2$ if $N = 200 \log(2^{d+2}/\delta)/\varepsilon^2$.*

*Proof.* $\rho$ is a quantum state so it is a Hermitian matrix positive semi definite of trace 1. Hence, we can write $\rho = \sum_{i=1}^{d} \lambda_i |\phi_i\rangle\langle\phi_i|$ where $\{\lambda_i\}_i$ is a probability distribution and $\{\phi_i\}_i$ is an orthonormal basis. Therefore $\sum_{i=1}^{d} |\phi_i\rangle\langle\phi_i| = \mathbb{I}$ and $\mathcal{M}$ is a valid POVM. Measuring $\rho$ via the POVM $\mathcal{M}$ is equivalent to sampling from the distribution $\{\mathrm{tr}(|\phi_i\rangle\langle\phi_i|\rho)\}_i = \{\sum_j \lambda_j \mathrm{tr}(|\phi_i\rangle\langle\phi_i||\phi_j\rangle\langle\phi_j|)\}_i = \{\lambda_i\}_i$ hence

$$A_1, \ldots, A_N \underset{i.i.d.}{\sim} \{\lambda_i\}_i.$$

On the other hand $\rho$ and $\tilde{\rho}$ have the same basis of diagonalization so the 1 norm between them is simply

$$\|\rho - \tilde{\rho}\|_{\mathrm{tr}} = \left\| \sum_{i=1}^{d} \lambda_i |\phi_i\rangle\langle\phi_i| - \sum_{i=1}^{d} \tilde{\lambda}_i |\phi_i\rangle\langle\phi_i| \right\|_{\mathrm{tr}}$$

$$= \left\| \sum_{i=1}^{d} (\lambda_i - \tilde{\lambda}_i) |\phi_i\rangle\langle\phi_i| \right\|_{\mathrm{tr}}$$

$$= \sum_{i=1}^{d} |\lambda_i - \tilde{\lambda}_i|$$

$$= 2 \, \mathrm{TV}(\lambda, \tilde{\lambda}),$$

where $\{\tilde{\lambda}_i\}_i = \{\frac{\sum_{j=1}^{N} \mathbf{1}_{A_j = i}}{N}\}_i$. It is well known that the TV distance can be written as:

$$\mathrm{TV}(\lambda, \tilde{\lambda}) = \max_{B \subset [d]} (\tilde{\lambda}(B) - \lambda(B)).$$

Chernoff-Hoeffding((Hoeffding, 1963)) inequality implies for all $B \subset [d]$ :

$$\mathbb{P}\left( \left| \frac{\sum_{j=1}^{N} \mathbf{1}_{A_j \in B}}{N} - \lambda(B) \right| > \frac{\varepsilon}{20} \right) \leq 2 \exp\left( -2N \left( \frac{\varepsilon}{20} \right)^2 \right).$$

Therefore by union bound we obtain

$$\mathbb{P}\left( \|\rho - \tilde{\rho}\|_{\mathrm{tr}} > \varepsilon/10 \right) = \mathbb{P}\left( 2 \, \mathrm{TV}(\lambda, \tilde{\lambda}) > \varepsilon/10 \right)$$

$$= \mathbb{P}\left( \exists B \subset [d] : \left| \frac{\sum_{j=1}^{N} \mathbf{1}_{A_j \in B}}{N} - \lambda(B) \right| > \frac{\varepsilon}{20} \right)$$

$$\leq 2^{d+1} \exp\left( -2N \left( \frac{\varepsilon}{20} \right)^2 \right).$$

Finally for $N = 200 \log(2^{d+1}/\delta)/\varepsilon^2$, we have with at least a probability $1 - \delta$ : $\|\rho - \tilde{\rho}\|_{\mathrm{tr}} \leq \varepsilon/10$. $\square$

Grouping the two previous Lemmas, Alg. 1 finds the closest quantum state $\sigma_{i^\star}$ and returns an $\varepsilon/10$-approximation of $\rho$ with a probability at least $1 - (\delta/2 + \delta/2) = 1 - \delta$. Finally, Alg. 1 is $\delta$-correct.

## C   Lower bound for the problem $(P)$

In this section, we focus on proving lower bounds for hypothesis selection problem $(P)$ for non-adaptive and adaptive strategies.

### C.1   Non-adaptive strategies

We recall the theorem we want to prove:

**Theorem C.1.** *There is a tuple of quantum states $(\sigma_1, \ldots, \sigma_m)$ such that any learning algorithm for problem (P) with non-adaptive incoherent measurements requires*

$$N = \Omega\left(\min\left\{\frac{md}{\log(m)\varepsilon^2}, \frac{d^2}{\varepsilon^2}\right\}\right)$$

*copies of $\rho$.*

**Construction**   For the construction, we choose $m$ unitary matrices $\{U_y\}_y$ chosen randomly from the $\mathrm{Haar}(d)$ distribution, then we choose for each unitary (orthonormal basis) random eigenvalues:

**Lemma C.2.** *Let $m \leq \exp(d^2/3000)$. Let $\{U_y\}_{y\in[m/2]}$ $m/2$ unitaries $\mathrm{Haar}(d)$ distributed. For $y \in [m/2]$, let $\sigma_y = 2\mathbb{I}/d - \sigma_{m+1-y} = U_y \Lambda U_y^\dagger$ where $\Lambda = \frac{\mathbb{I}}{d} + \mathrm{diag}\left(\{\lambda_i\}_{i\in[d]}\right) = \mathrm{diag}\left(\left\{\frac{1+(-1)^d 10\varepsilon}{d}\right\}_{i\in[d]}\right)$. We have with a probability at least $9/10$, for all $y \neq z \in [m]$:*

$$\|\sigma_y - \sigma_z\|_{\mathrm{tr}} \geq \varepsilon.$$

*Proof.* Let $y \neq z \in [m/2]$ and $0 \preccurlyeq O \preccurlyeq \mathbb{I}$ satisfying $\mathrm{tr}\,\mathrm{diag}\left(\{\lambda_i\}_{i\in[d]}\right)O = -5\varepsilon$. Let $f(U) = \mathrm{tr}\left(U\,\mathrm{diag}\left(\{\lambda_i\}_{i\in[d]}\right)U^\dagger - \mathrm{diag}\left(\{\lambda_i\}_{i\in[d]}\right)\right)O$ where $U \sim \mathrm{Haar}(d)$, we have $\mathbb{E}(f(U)) = -\mathrm{tr}\,\mathrm{diag}\left(\{\lambda_i\}_{i\in[d]}\right)O = 5\varepsilon$ (see Weingarten Calculus D.1). The function $f$ is $\frac{20\varepsilon}{\sqrt{d}}$-Lipschitz:

$$\begin{aligned}
&|f(U) - f(V)| \\
&= |\mathrm{tr}(U\,\mathrm{diag}\left(\{\lambda_i\}_{i\in[d]}\right)U^\dagger - \mathrm{diag}\left(\{\lambda_i\}_{i\in[d]}\right))O - \mathrm{tr}(V\,\mathrm{diag}\left(\{\lambda_i\}_{i\in[d]}\right)V^\dagger - \mathrm{diag}\left(\{\lambda_i\}_{i\in[d]}\right))O| \\
&\leq |\mathrm{tr}(U\,\mathrm{diag}\left(\{\lambda_i\}_{i\in[d]}\right)U^\dagger - V\,\mathrm{diag}\left(\{\lambda_i\}_{i\in[d]}\right)V^\dagger)O| \\
&\leq \|(U-V)\,\mathrm{diag}\left(\{\lambda_i\}_{i\in[d]}\right)U^\dagger\|_{\mathrm{tr}} + \|V\,\mathrm{diag}\left(\{\lambda_i\}_{i\in[d]}\right)(U-V)^\dagger\|_{\mathrm{tr}} \\
&\leq \|U-V\|_2\|\mathrm{diag}\left(\{\lambda_i\}_{i\in[d]}\right)U^\dagger\|_2 + \|V\,\mathrm{diag}\left(\{\lambda_i\}_{i\in[d]}\right)\|_2\|(U-V)^\dagger\|_2 \\
&\leq \frac{10\varepsilon}{d}(\|U-V\|_2\|\mathrm{diag}\left(\{(-1)^i\}_{i\in[d]}\right)U^\dagger\|_2 + \|V\,\mathrm{diag}\left(\{(-1)^d\}_{i\in[d]}\right)\|_2\|(U-V)^\dagger\|_2) \\
&\leq \frac{20\varepsilon}{\sqrt{d}}\|U-V\|_2,
\end{aligned}$$

where we have used Cauchy Schwarz inequality.

Using the fact that the Haar distribution is invariant under the multiplication by a unitary and the concentration inequality for Lipschitz functions D.3, the probability that the states $\{\sigma_y\}_{y\in[m/2]}$ are not $\varepsilon$-separated

is upper bounded by

$$
\begin{aligned}
\mathbb{P}\left(\exists y,z \in [m/2] : \|\sigma_y - \sigma_z\|_{\mathrm{tr}} \le \varepsilon\right) &\le \frac{m^2}{4}\mathbb{P}\left(\|\sigma_y - \sigma_z\|_{\mathrm{tr}} \le \varepsilon\right) \\
&\le \frac{m^2}{4}\mathbb{P}\left(\|U_y \operatorname{diag}\left(\{\lambda_i\}_{i\in[d]}\right)U_y^\dagger - U_z \operatorname{diag}\left(\{\lambda_i\}_{i\in[d]}\right)U_z^\dagger\|_{\mathrm{tr}} \le \varepsilon\right) \\
&\le \frac{m^2}{4}\mathbb{P}\left(\|U_z^\dagger U_y \operatorname{diag}\left(\{\lambda_i\}_{i\in[d]}\right)U_y^\dagger U_z - \operatorname{diag}\left(\{\lambda_i\}_{i\in[d]}\right)\|_{\mathrm{tr}} \le \varepsilon\right) \\
&\le \frac{m^2}{4}\mathbb{P}\left(\|U \operatorname{diag}\left(\{\lambda_i\}_{i\in[d]}\right)U^\dagger - \operatorname{diag}\left(\{\lambda_i\}_{i\in[d]}\right)\|_{\mathrm{tr}} \le \varepsilon\right) \\
&\le \frac{m^2}{4}\mathbb{P}\left(\operatorname{tr}\left(U \operatorname{diag}\left(\{\lambda_i\}_{i\in[d]}\right)U^\dagger - \operatorname{diag}\left(\{\lambda_i\}_{i\in[d]}\right)\right)O \le \varepsilon\right) \\
&\le \frac{m^2}{4}\mathbb{P}\left(f(U) - \mathbb{E}\left(f(U)\right) \le \varepsilon - 5\varepsilon\right) \\
&\le \frac{m^2}{4}\mathbb{P}\left(\mathbb{E}\left(f(U)\right) - f(U) \ge 4\varepsilon\right) \\
&\le \frac{m^2}{4}\exp\left(-\frac{16d^2\varepsilon^2}{12\times 400\varepsilon^2}\right) \\
&\le \frac{m^2}{4}\exp\left(-\frac{d^2}{1000}\right),
\end{aligned}
$$

which is smaller than $1/10$ if $m^2 \le 2\exp(d^2/1000)/5$.

For the case when $y \in [m/2]$ and $z \in [m]\setminus[m/2]$, let $x = m+1-z \in [m/2]$ we have

$$
\begin{aligned}
\|\sigma_y - \sigma_z\|_{\mathrm{tr}} &= \|\sigma_y - 2\mathbb{I}/d + \sigma_x\|_{\mathrm{tr}} \\
&\ge \|\sigma_x - 2\mathbb{I}/d + \sigma_x\|_{\mathrm{tr}} - \|\sigma_y - \sigma_x\|_{\mathrm{tr}} \\
&\ge 2\|\sigma_x - \mathbb{I}/d\|_{\mathrm{tr}} + \|_{\mathrm{tr}} - \|\sigma_y - \sigma_x\|_{\mathrm{tr}} \\
&\ge \varepsilon.
\end{aligned}
$$

Finally, for the case when $y \in [m]\setminus[m/2]$ and $z \in [m]\setminus[m/2]$, let $y' = m+1-y \in [m/2]$ and $z' = m+1-z \in [m/2]$ we have

$$
\begin{aligned}
\|\sigma_y - \sigma_z\|_{\mathrm{tr}} &= \|2\mathbb{I}/d - \sigma_{y'} - 2\mathbb{I}/d + \sigma_{z'}\|_{\mathrm{tr}} \\
&\ge \|\sigma_{y'} - \sigma_{z'}\|_{\mathrm{tr}} \\
&\ge \varepsilon.
\end{aligned}
$$

$\square$

We have shown how to construct the unitaries, we move to prove the existence of the eigenvalues:

**Lemma C.3.** *There exists family of quantum states $\{\rho_{x,y}\}_{|x|\in[e^{cd}],y\in[m]}$ (where $c$ is a universal constant) such that for each $y \in [m]$, $\{\rho_{x,y}\}_{|x|\in[e^{cd}]}$ is $\varepsilon/5$-separated and commute.*

*Proof.* We start by writing the eigen-decomposition of the known quantum states $\sigma_y$ as

$$
\sigma_y = U_y\left(\sum_{i=1}^d \lambda_i^y |i\rangle\langle i|\right)U_y^\dagger.
$$

We claim that we can choose $\alpha_i^x$ to construct an $\varepsilon/5$-separated family of $me^{cd}$ quantum states ($c$ is a constant to be chosen later) of the form

$$
\rho_{x,y} = U_y\left(\sum_{i=1}^d \left(\lambda_i^y + \frac{\alpha_i^x(2\varepsilon/3)}{d}\right)|i\rangle\langle i|\right)U_y^\dagger,
$$

for $|x| \in [e^{cd}]$ and $y \in [m]$. Note that for convenience of notation, the labels $x$ can be positive and negative. Moreover the distance between $\rho_{x,y}$ and $\sigma_y$ is exactly:

$$\|\rho_{x,y} - \sigma_y\|_{\mathrm{tr}} = \frac{\varepsilon}{3}.$$

Concretely, we look for $\{\alpha_i^x\}_{1 \le i \le d, 1 \le |x| \le e^{cd}/2}$ such that

1. $\alpha_i^x = \pm 1$,

2. $\alpha_i^{-x} = -\alpha_i^x$,

3. $\alpha_i^x + \alpha_{i+d/2}^x = 0$ (we suppose $d$ is even) and

4. $\forall x \ne x' : \sum_{i=1}^{d/2} |\alpha_i^x - \alpha_i^{x'}| > d(1/2 - 1/200)$.

The third point ensures that $\rho$ has trace 1 while the fourth one implies $\|\rho_{x,y} - \rho_{x',y}\|_{\mathrm{tr}} > \varepsilon/3 - \varepsilon/100 > \varepsilon/5$. Starting by the simple quantum states $\rho_{1,y} = \sigma_y + \sum_{i=1}^{d/2} \frac{(2\varepsilon/3)}{d} U_y |i\rangle\langle i| U_y^\dagger - \sum_{i=d/2+1}^{d} \frac{(2\varepsilon/3)}{d} U_y |i\rangle\langle i| U_y^\dagger$ and $\rho_{-1,y} = 2\mathbb{I}/d - \rho_{1,y} = \sigma_{m+1-y} - \sum_{i=1}^{d/2} \frac{(2\varepsilon/3)}{d} U_y |i\rangle\langle i| U_y^\dagger + \sum_{i=d/2+1}^{d} \frac{(2\varepsilon/3)}{d} U_y |i\rangle\langle i| U_y^\dagger$ and we suppose that we have constructed $\mathcal{Q}$ an $\varepsilon$-separated family of the form described above of cardinality $M < e^{cd}$. Let $\alpha_1, \ldots, \alpha_{d/2}$ i.i.d. random variables taken values in $\{\pm 1\}$ with probability $1/2$ each. We have by Hoeffding's inequality

$$\mathbb{P}\left(\exists \rho_x \in \mathcal{Q} : \sum_{i=1}^{d/2} |\alpha_i^x - \alpha_i| \le d(1/2 - 1/200) \text{ OR } \sum_{i=1}^{d/2} |\alpha_i^{-x} - \alpha_i| \le d(1/2 - 1/200)\right)$$

$$= \mathbb{P}\left(\exists \rho_x \in \mathcal{Q} : \sum_{i=1}^{d/2} |\alpha_i^x - \alpha_i| \le d(1/2 - 1/200) \text{ OR } \sum_{i=1}^{d/2} |\alpha_i^x + \alpha_i| \le d(1/2 - 1/200)\right)$$

$$\le \frac{M}{2} \mathbb{P}\left(\sum_{i=1}^{d/2} \mathbf{1}_{\alpha_i = \alpha_i^x} > d/4 + d/400\right) + \frac{M}{2} \mathbb{P}\left(\sum_{i=1}^{d/2} \mathbf{1}_{\alpha_i = \alpha_i^x} \le d/4 - d/400\right)$$

$$\le M e^{-d/2000},$$

which is strictly less than 1 if $M < e^{d/2000}$. So let's take $c = 1/2000$, we deduce that

$$\mathbb{P}\left(\forall \rho_x \in \mathcal{Q} : \sum_{i=1}^{d/2} |\alpha_i^x - \alpha_i| > d(1/2 - 1/200)\right) > 0.$$

therefore there exists some $\alpha \in \{\pm 1\}^d$ verifying the desired conditions. We can repeat this construction until $Card(\mathcal{Q}) \ge e^{cd}$.

$\square$

We have constructed the $\varepsilon$-separated family of quantum states $\{\sigma_y\}_y$ an the corresponding $\varepsilon/5$-separated $\{\rho_{x,y}\}_x$ for all $y$, we can use tools from communication theory to deduce the lower bound (see (Haah et al., 2016)). Alice encodes a message $(x, y) \in \{1, \ldots, e^{cd}\} \times [m]$ in $\rho_{x,y}$ and sends it to Bob. To read the message, Bob tries to approximate the quantum state that he received from Alice. We suppose that Bob can approximate (up to $\varepsilon/10$ in trace norm) a state $\varepsilon/3$ close to one of $\{\sigma_y\}$ and diagonalized in the same basis of this quantum state with a probability at least $2/3$. Bob uses $N$ copies to decode Alice's message and returns $(x', y') \in \{1, \ldots, e^{cd}\} \times [m]$, therefore by Fano's inequality ((Fano, 1961)) we have the following lower bound on the mutual information:

**Lemma C.4** (Fano). *The mutual information can be lower bounded:*

$$I(X, Y : X', Y') \ge 2/3 \log(m e^{cd}) - \log(2) \ge \Omega(\log(m) + d).$$

On the other hand we can upper bound the mutual information between $(X, Y)$ and $(X', Y')$. Let $I_1, \ldots, I_N$ be the outcomes of a non adaptive algorithm solving the problem $(P)$. By using the data-processing inequality for the Kullback-Leibler divergence and the fact that every non adaptive algorithm for the problem $(P)$ can be used as a 2/3-correct decoder we can upper bound the mutual information as follows:

**Lemma C.5** (Data-processing). *The mutual information between $(X, Y)$ and $(X', Y')$ is smaller than the mutual information between $(X, Y)$ and $(I_1, \ldots, I_N)$:*

$$I(X, Y : X', Y') \leq I(X, Y : I_1, \ldots, I_N).$$

The next step is to upper bound the mutual information between $(X, Y)$ and $(I_1, \ldots, I_N)$. This latter depends on the quantum states $\{\sigma_y\}_y$, therefore it is a random variable. We will show that with at least a probability 9/10, it is upper bounded by an expression involving the parameters of the problem. First we start by proving the following upper bound relating the mutual information with the unitaries $\{U_y\}_y$ defining the quantum states $\{\sigma_y\}_y$.

**Lemma C.6.** *For all unitaries $\{U_y\}_y$, we have:*

$$I(X, Y : I_1, \ldots, I_N) \leq 4N \sup_{\phi, \|\phi\|_2 \leq 1} \frac{1}{M} \sum_{|x|, y \leq m/2} \langle \phi | U_y O_{x,y} U_y^\dagger | \phi \rangle^2 \varepsilon^2,$$

*where for $(x, y)$, $O_{x,y} = U_y^\dagger (d\rho_{x,y} - \mathbb{I}) U_y$.*

*Proof.* We suppose that the eigenvalues of $\sigma_y$ have the form

$$\lambda_i^y = \frac{1 + 10\beta_i^y \varepsilon}{d},$$

where $\beta_i^y = \pm 1$ satisfying $\sum_i \beta_i^y = 0$ (exactly half are equal to $+1$) and $\beta^y = -\beta^{m+1-y}$ (we suppose $m$ even). The diagonalizing matrices $\{U_y\}_y$ are chosen randomly so as they satisfy $U_{m+1-y} = U_y$ for all $y \leq m/2$ and other conditions to be specified later.

Let us denote by $\mathcal{M}^t$ the POVM used at step $t$. Without loss of generality, we can suppose that the non-adaptive algorithm performs only measurements of the following form:

$$\mathcal{M}^t = \{|\phi_i^t\rangle\langle\phi_i^t|\}_i.$$

where we have the condition $\sum_i |\phi_i^t\rangle\langle\phi_i^t| = \mathbb{I}$ implying for all $i$ and $t$: $\|\phi_i^t\|_2 \leq 1$.

Let $M = 2me^{cd}$, we can write the mutual information as follows:

$$I(X, Y : I_1, \ldots, I_N) = H\left(\frac{1}{M}\sum_{x,y} \operatorname{tr}((\rho_{x,y})^{\otimes N} \otimes_{t=1}^N \mathcal{M}^t)\right) - \frac{1}{M}\sum_{x,y} H\left(\operatorname{tr}((\rho_{x,y})^{\otimes N} \otimes_{t=1}^N \mathcal{M}^t)\right)$$

$$= \frac{1}{M}\sum_{x,y}\sum_{i_1,\ldots,i_N}\prod_{t=1}^N \langle\phi_{i_t}^t|\rho_{x,y}|\phi_{i_t}^t\rangle \log\left(\frac{\prod_{t=1}^N \langle\phi_{i_t}^t|\rho_{x,y}|\phi_{i_t}^t\rangle}{\frac{1}{M}\sum_{x,y}\prod_{t=1}^N \langle\phi_{i_t}^t|\rho_{x,y}|\phi_{i_t}^t\rangle}\right) = \Sigma_1 + \Sigma_2,$$

where $\Sigma_1$ and $\Sigma_2$ are defined as follows:

$$\Sigma_1 = \frac{1}{M}\sum_{x,y}\sum_{i_1,\ldots,i_N}\prod_{t=1}^N \langle\phi_{i_t}^t|\rho_{x,y}|\phi_{i_t}^t\rangle \log\left(\prod_{t=1}^N \langle\phi_{i_t}^t|d\rho_{x,y}|\phi_{i_t}^t\rangle\right),$$

$$\Sigma_2 = -\frac{1}{M}\sum_{x,y}\sum_{i_1,\ldots,i_N}\prod_{t=1}^N \langle\phi_{i_t}^t|\rho_{x,y}|\phi_{i_t}^t\rangle \log\left(\frac{1}{M}\sum_{x,y}\prod_{t=1}^N \langle\phi_{i_t}^t|d\rho_{x,y}|\phi_{i_t}^t\rangle\right).$$

Since

$$\rho_{x,y} = U_y \operatorname{diag}\left(\left\{\frac{1 + (10\beta_i^y + 2\alpha_i^x/3)\varepsilon}{d}\right\}_{i\in[d]}\right) U_y^\dagger$$

we can write

$$\langle \phi_{i_t}^t | \rho_{x,y} | \phi_{i_t}^t \rangle = \frac{1 + u_{i_t}^{t,x,y}\varepsilon}{d},$$

where $u_{i_t}^{t,x,y} = \langle \phi_{i_t}^t | U_y \operatorname{diag}\left(\{10\beta_i^y + 2\alpha_i^x/3\}_{i\in[d]}\right) U_y | \phi_{i_t}^t \rangle \in (-11, 11)$. Denote by $O_{x,y} = \operatorname{diag}\left(\{10\beta_i^y + 2\alpha_i^x/3\}_{i\in[d]}\right)$, we remark that

$$\sum_{i_t=1}^d u_{i_t}^{t,x,y} = \sum_{i_t=1}^d \langle \phi_{i_t}^t | U_y \operatorname{diag}\left(\{10\beta_i^y + 2\alpha_i^x/3\}_{i\in[d]}\right) U_y | \phi_{i_t}^t \rangle = \operatorname{tr} U_y \operatorname{diag}\left(\{10\beta_i^y + 2\alpha_i^x/3\}_{i\in[d]}\right) U_y$$

$$= \operatorname{tr} \operatorname{diag}\left(\{10\beta_i^y + 2\alpha_i^x/3\}_{i\in[d]}\right) = \sum_{i=1}^d 10\beta_i^y + 2\alpha_i^x/3 = 0.$$

Moreover, the couples of quantum states $(\rho_{x,y}, \rho_{-x,y})$ and $(\rho_{x,y}, \rho_{x,m+1-y})$ are symmetric with respect to $\mathbb{I}/d$ by the construction of $(\alpha_i^x)_{i,x}$ and $(\beta_i^x)_{i,x}$ hence

$$u_{i_t}^{t,-x,m+1-y} = \langle \phi_{i_t}^t | U_{m+1-y} \operatorname{diag}\left(\left\{10\beta_i^{m+1-y} + 2\alpha_i^{-x}/3\right\}_{i\in[d]}\right) U_{m+1-y} | \phi_{i_t}^t \rangle$$

$$= \langle \phi_{i_t}^t | U_y \operatorname{diag}\left(\{-10\beta_i^y - 2\alpha_i^x/3\}_{i\in[d]}\right) U_y | \phi_{i_t}^t \rangle$$

$$= -\langle \phi_{i_t}^t | U_y \operatorname{diag}\left(\{10\beta_i^y + 2\alpha_i^x/3\}_{i\in[d]}\right) U_y | \phi_{i_t}^t \rangle$$

$$= -u_{i_t}^{t,x,y}.$$

Suppose that $\varepsilon \leq 0.05$. We can start by controlling $\Sigma_2$ using Jensen's inequality:

$$\Sigma_2 = -\frac{1}{M}\sum_{x,y}\sum_{i_1,\dots,i_N}\prod_{t=1}^N\left(\frac{1 + u_{i_t}^{t,x,y}\varepsilon}{d}\right)\log\left(\frac{1}{M}\sum_{x,y}\prod_{t=1}^N(1 + u_{i_t}^{t,x,y}\varepsilon)\right)$$

$$\leq -\frac{1}{M}\sum_{x,y,i}\prod_{t=1}^N\left(\frac{1 + u_{i_t}^{t,x,y}\varepsilon}{d}\right)\left(\frac{1}{M}\sum_{x,y}\log\left(\prod_{t=1}^N(1 + u_{i_t}^{t,x,y}\varepsilon)\right)\right)$$

$$= -\frac{1}{M}\sum_{x,y,i}\prod_{t=1}^N\left(\frac{1 + u_{i_t}^{t,x,y}\varepsilon}{d}\right)\left(\frac{1}{M}\sum_{x,y,t}\log\left(1 + u_{i_t}^{t,x,y}\varepsilon\right)\right)$$

$$= -\frac{1}{M}\sum_{x,y,i}\prod_{t=1}^N\left(\frac{1 + u_{i_t}^{t,x,y}\varepsilon}{d}\right)\left(\frac{1}{M}\sum_{|x|,y\leq m/2,t}\log\left(1 + u_{i_t}^{t,x,y}\varepsilon\right) + \log\left(1 - u_{i_t}^{t,x,y}\varepsilon\right)\right)$$

$$= -\frac{1}{M}\sum_{x,y,i}\prod_{t=1}^N\left(\frac{1 + u_{i_t}^{t,x,y}\varepsilon}{d}\right)\left(\frac{1}{M}\sum_{|x|,y\leq m/2,t}\log\left(1 - (u_{i_t}^{t,x,y})^2\varepsilon^2\right)\right).$$

Now, we can use the inequality $-\log(1 - x^2) \leq 2x^2$ for $|x| \leq 1/\sqrt{2}$:

$$
\begin{aligned}
\Sigma_2 &\leq -\frac{1}{M} \sum_{x,y,i} \prod_{t=1}^{N} \left( \frac{1 + u_{i_t}^{t,x,y} \varepsilon}{d} \right) \left( \frac{1}{M} \sum_{|x|,y \leq m/2,t} \log\left(1 - (u_{i_t}^{t,x,y})^2 \varepsilon^2\right) \right) \\
&\leq \frac{1}{M} \sum_{x,y,i} \prod_{t=1}^{N} \left( \frac{1 + u_{i_t}^{t,x,y} \varepsilon}{d} \right) \left( \frac{1}{M} \sum_{|x|,y \leq m/2,t} 2(u_{i_t}^{t,x,y})^2 \varepsilon^2 \right) \\
&\leq \frac{1}{M} \sum_{x,y,i} \prod_{t=1}^{N} \left( \frac{1 + u_{i_t}^{t,x,y} \varepsilon}{d} \right) \left( \sum_{t} \sup_{\phi, \|\phi\|_2 \leq 1} \frac{1}{M} \sum_{|x|,y \leq m/2} 2\langle \phi | U_y O_{x,y} U_y^\dagger | \phi \rangle^2 \varepsilon^2 \right) \\
&\leq N \sup_{\phi, \|\phi\|_2 \leq 1} \frac{1}{M} \sum_{|x|,y \leq m/2} 2\langle \phi | U_y O_{x,y} U_y^\dagger | \phi \rangle^2 \varepsilon^2.
\end{aligned}
$$

Using the fact that $\sum_{i_t} u_{i_t}^{t,x,y} = 0$ for all $t,x,y$ along with the inequality $(1+x)\log(1+x) + (1-x)\log(1-x) \leq 2x^2$ for $|x| \leq 1/\sqrt{2}$ we can upper bound the first sum $\Sigma_1$:

$$
\begin{aligned}
\Sigma_1 &= \frac{1}{M} \sum_{x,y,i} \prod_{t=1}^{N} \left( \frac{1 + u_{i_t}^{t,x,y} \varepsilon}{d} \right) \log\left( \prod_{t=1}^{N} \left(1 + u_{i_t}^{t,x,y} \varepsilon\right) \right) \\
&\leq \frac{1}{M} \sum_{x,y,i} \prod_{t=1}^{N} \left( \frac{1 + u_{i_t}^{t,x,y} \varepsilon}{d} \right) \sum_{k} \log\left(1 + u_{i_k}^{k,x,y} \varepsilon\right) \\
&\leq \frac{1}{M} \sum_{x,y,k} \sum_{i_k} \sum_{i_1,\dots,i_{k-1},i_{k+1},\dots,i_N} \prod_{t=1}^{N} \left( \frac{1 + u_{i_t}^{t,x,y} \varepsilon}{d} \right) \log\left(1 + u_{i_k}^{k,x,y} \varepsilon\right) \\
&\leq \frac{1}{M} \sum_{x,y,k} \sum_{i_k} \left( \frac{1 + u_{i_k}^{k,x,y} \varepsilon}{d} \right) \log\left(1 + u_{i_k}^{k,x,y} \varepsilon\right) \\
&\leq \frac{1}{Md} \sum_{|x|,y \leq m/2,k} \sum_{i_k} \left(1 + u_{i_k}^{k,x,y} \varepsilon\right) \log\left(1 + u_{i_k}^{k,x,y} \varepsilon\right) + \left(1 - u_{i_k}^{k,x,y} \varepsilon\right) \log\left(1 - u_{i_k}^{k,x,y} \varepsilon\right) \\
&\leq \frac{1}{Md} \sum_{|x|,y \leq m/2,k,i_k} 2(u_{i_k}^{k,x,y} \varepsilon)^2 \\
&\leq \frac{1}{d} \sum_{k,i_k} \sup_{\phi, \|\phi\|_2 \leq 1} \frac{1}{M} \sum_{|x|,y \leq m/2} 2\langle \phi | U_y O_{x,y} U_y^\dagger | \phi \rangle^2 \varepsilon^2 \\
&\leq 2N \sup_{\phi, \|\phi\|_2 \leq 1} \frac{1}{M} \sum_{|x|,y \leq m/2} \langle \phi | U_y O_{x,y} U_y^\dagger | \phi \rangle^2 \varepsilon^2.
\end{aligned}
$$

Finally the upper bounds on $\Sigma_1$ and $\Sigma_2$ imply the required upper bound on their sum $\Sigma_1 + \Sigma_2 = I(X, Y : I_1, \dots, I_N)$. $\qquad\square$

Note that we need to take a supremum over all possible vectors $\phi$ because the learner knows the quantum states $\{\sigma_y\}_y$ and so it can choose measurements dependent on the unitaries $\{U_y\}_y$. We can now show that with high probability on the choice of the unitaries $\{U_y\}_y$, the latter supremum can bounded and so the mutual information too.

**Lemma C.7.** *Let $\{U_y\}_y$ $m$ unitary matrices $\mathrm{Haar}(d)$ distributed. We have with a probability at least $9/10$:*

$$
4N \sup_{\phi, \|\phi\|_2 \leq 1} \frac{1}{M} \sum_{|x|,y \leq m/2} \langle \phi | U_y O_{x,y} U_y^\dagger | \phi \rangle^2 \varepsilon^2 = \mathcal{O}\left( \frac{N\varepsilon^2 \log(m)}{m} + \frac{N\varepsilon^2}{d} \right).
$$

*Proof.* To upper bound the previous supremum, we use a similar approach to (Chen et al., 2021): For $U \sim$ Haar$(d)$, $\phi \in B(0,1)$ and a trace-less Hermitian matrix $O$, let $f(\phi, U) = \langle \phi | UOU^\dagger | \phi \rangle$, we have $\mathbb{E}\left(f(\phi, U)\right) = \frac{1}{d}\text{tr}(O)\text{tr}(|\phi\rangle\langle\phi|) = 0$ (see Weingarten Calculus D.1) and $f$ is $2\|O\|$-Lipschitz:

$$|f(U) - f(V)| \leq 2|\langle \phi |(U - V)OU^\dagger|\phi\rangle| \leq 2\|O\|\|U - V\|_2.$$

Therefore by the concentration inequality D.3:

$$\mathbb{P}\left(|f(U)| > t\right) \leq \exp(-dt^2/48).$$

Hence

$$\mathbb{P}\left(|f(U)|^2 > t\right) \leq \exp(-dt/48).$$

For $m/2$ unitaries $U_1, \ldots, U_{m/2}$ and $\lambda = 2d/C$ for sufficiently large $C$. Denote by $X = |f(U)|^2$, by Markov's inequality:

$$\mathbb{P}\left(\frac{2}{m}\sum_{1 \leq y \leq m/2} |f(U_y)|^2 > t\right) \leq \exp(-\lambda mt/2)\mathbb{E}\left(e^{\lambda X}\right)^{m/2} \leq \exp(-\lambda mt/2)\left(1 + \int_0^\infty dx\,\lambda e^{\lambda x}e^{-dx/48}\right)^{m/2}$$

$$\leq \exp(-dmt/2C)\left(C'\right)^{m/2} \leq \exp(-dmt/C + m\log(C')),$$

with $C'$ another constant. In order to prove an inequality valid for all $\phi \in B(0,1)$, let's take an $\eta$-net $\{\phi_i\}_i$ of size at most $(1 + 2/\eta)^{2d}$. For $\phi \in B(0,1)$, there is $\phi_i$ such that $\|\phi - \phi_i\|_2 \leq \eta$. Moreover $|f(\phi, U)| \leq \|O\|$ so

$$\left|\frac{2}{m}\sum_{1 \leq y \leq m/2} f(\phi, U_y)^2 - f(\phi_i, U_y)^2\right| \leq \frac{2}{m}\sum_{1 \leq y \leq m/2} |f(\phi, U_y)^2 - f(\phi_i, U_y)^2|$$

$$\leq \frac{2}{m}\sum_{1 \leq y \leq m/2} 2\|O\||(\langle\phi| - \langle\phi_i|)U_yOU_y^\dagger|\phi\rangle| \leq 2\eta\|O\|^2.$$

Therefore

$$\mathbb{P}\left(\exists \phi : \frac{2}{m}\sum_{1 \leq y \leq m/2} |f(\phi, U_y)|^2 > t + 2\eta\|O\|^2\right) \leq \mathbb{P}\left(\exists \phi_i : \frac{1}{m}\sum_{k=1}^m |f(\phi_i, U_k)|^2 > t\right)$$

$$\leq (1 + 2/\eta)^{2d}\exp(-dmt/C + m\log(C')).$$

Taking $\eta = 1/m$ yields:

$$\mathbb{P}\left(\exists \phi : \frac{2}{m}\sum_{1 \leq y \leq m/2} |f(\phi, U_y)|^2 > t + 2\|O\|^2/m\right) \leq (1 + 2m)^{2d}\exp(-dmt/C + m\log(C')).$$

Applying the union bound, we can obtain:

$$\mathbb{P}\left(\exists \phi, \exists x, \frac{2}{m}\sum_{y \leq m/2} \langle\phi|U_yO_{x,y}U_y^\dagger|\phi\rangle^2 \geq t + \frac{2\|O_{x,y}\|^2}{m}\right) \leq 4e^{cd}(1 + 2m)^{2d}\exp(-dmt/C + m\log(C')).$$

Let's take $t = C\frac{\log(40) + cd + 2d\log(1+2m) + m\log(C')}{dm}$ in order to have

$$\mathbb{P}\left(\forall\phi, \frac{1}{M}\sum_{|x|, y \leq m/2} \langle\phi|U_yO_{x,y}U_y^\dagger|\phi\rangle^2 \leq t + \frac{2\|O_{x,y}\|^2}{m}\right)$$

$$\geq \mathbb{P}\left(\forall\phi, \forall x, \frac{1}{m}\sum_{y \leq m/2} \langle\phi|U_yO_{x,y}U_y^\dagger|\phi\rangle^2 \leq t + \frac{2\|O_{x,y}\|^2}{m}\right)$$

$$\geq 9/10.$$

Therefore we have the existence of $\{U_y\}_y$ such that for all $y \neq z$

$$\|\sigma_y - \sigma_z\|_{\mathrm{tr}} > \varepsilon,$$

and

$$\sup_{\phi, \|\phi\|_2 \leq 1} \frac{1}{M} \sum_{|x|, y \leq m/2} \langle \phi | U_y O_{x,y} U_y^\dagger | \phi \rangle^2 \leq \frac{\mathrm{tr}(O_{x,y}^2)}{d(d+1)} + t + \frac{2\|O_{x,y}\|^2}{m}$$

$$\leq \frac{201}{d+1} + C \frac{\log(40) + cd + 2d\log(1+2m) + m\log(C')}{dm} + \frac{242}{m}.$$

Finally, we showed the existence of quantum states $\{\sigma_{x,y}\}_{x,y}$ such that:

$$I(X, Y : I_1, \ldots, I_N) = \mathcal{O}\left(\frac{1}{d} + \frac{\log(m)}{m}\right) N\varepsilon^2.$$

$\square$

To sum up, we have shown the existence of quantum states $\{\sigma_{x,y}\}_{x,y}$ such that:

$$\Omega(\log(m) + d) \leq I(X, Y : X', Y') \leq I(X, Y : I_1, \ldots, I_N) \leq \mathcal{O}\left(\frac{1}{d} + \frac{\log(m)}{m}\right) N\varepsilon^2.$$

We conclude that $N = \Omega\left(\min\left\{\frac{md}{\log(m)\varepsilon^2}, \frac{d^2}{\varepsilon^2}\right\}\right)$.

## C.2   Adaptive strategies

It is important to see why this proof doesn't work for adaptive strategies. The lower bound on the mutual information has nothing to do with the non-adaptive/adaptive option of the algorithm so it remains true. However, upper bounding the mutual information cannot be done the same way since now the POVM used at time $t$ depends on the previous outcomes. Let $\{u_t\}_{t=1}^N$ be a sequence constituted by the outcomes of a correct adaptive algorithm. Let $\mathcal{M}_{u_{<t}}^t = \{|\phi_v^{u_{<t}}\rangle\langle\phi_v^{u_{<t}}|\}_v$ the POVM used at time $t$ given the previous outcomes $u_{<t}$. Recall that the mutual information between $(X, Y)$ and $(I_1, \ldots, I_N)$ can be expressed as:

$$I(X, Y : I_1, \ldots, I_N) = \Sigma_1 + \Sigma_2.$$

The second sum can be upper bounded by the same technique as before (using for example Jensen's inequality and the inequality $-\log(1 - x^2) \leq 2x^2$) and yields the same upper bound. The first sum is more involved because the product cannot be simplified due to the dependence between the POVMs and the previous outcomes. To see this, we can try to simplify the first sum as far as possible:

$$\Sigma_1 = \frac{1}{M} \sum_{x,y} \sum_{u_1, \ldots, u_N} \prod_{t=1}^N \langle \phi_{u_t}^{u_{<t}} | \rho_{x,y} | \phi_{u_t}^{u_{<t}} \rangle \log \left( \prod_{t=1}^N \langle \phi_{u_t}^{u_{<t}} | d\rho_{x,y} | \phi_{u_t}^{u_{<t}} \rangle \right)$$

$$= \frac{1}{M} \sum_{x,y} \sum_{u_1, \ldots, u_N} \prod_{t=1}^N \langle \phi_{u_t}^{u_{<t}} | \rho_{x,y} | \phi_{u_t}^{u_{<t}} \rangle \sum_{t=1}^N \log \left( \langle \phi_{u_t}^{u_{<t}} | d\rho_{x,y} | \phi_{u_t}^{u_{<t}} \rangle \right)$$

$$= \frac{1}{M} \sum_{x,y,k} \sum_{u_1, \ldots, u_N} \prod_{t=1}^N \langle \phi_{u_t}^{u_{<t}} | \rho_{x,y} | \phi_{u_t}^{u_{<t}} \rangle \log \left( \langle \phi_{u_k}^{u_{<k}} | d\rho_{x,y} | \phi_{u_k}^{u_{<k}} \rangle \right)$$

$$= \frac{1}{M} \sum_{x,y,k} \sum_{u_1, \ldots, u_k} \prod_{t=1}^k \langle \phi_{u_t}^{u_{<t}} | \rho_{x,y} | \phi_{u_t}^{u_{<t}} \rangle \log \left( \langle \phi_{u_k}^{u_{<k}} | d\rho_{x,y} | \phi_{u_k}^{u_{<k}} \rangle \right),$$

where the last equality follows from the fact that

$$\sum_{u_t} \langle \phi_{u_t}^{u_{<t}} | \rho_{x,y} | \phi_{u_t}^{u_{<t}} \rangle = \text{tr}(\rho_{x,y}) = 1,$$

for $t > k$ and $\log\left(\langle \phi_{u_k}^{u_{<k}} | d\rho_{x,y} | \phi_{u_k}^{u_{<k}} \rangle\right)$ is independent from $u_t$. But we are stuck at $k$, we cannot simplify the sums on $u_s$ for $s < k$ since $\langle \phi_{u_s}^{u_{<s}} | \rho_{x,y} | \phi_{u_s}^{u_{<s}} \rangle$ has common terms with $\langle \phi_{u_k}^{u_{<k}} | \rho_{x,y} | \phi_{u_k}^{u_{<k}} \rangle$ which is inside the log function.

In order to circumvent this difficulty, we can upper bound the $k^{th}$ term which poses the obstacle of simplification. Using the inequality $\log(x) \leq x - 1$ for all $x > -1$ we obtain:

$$\Sigma_1 = \frac{1}{M} \sum_{x,y,k} \sum_{u_1,\ldots,u_k} \prod_{t=1}^{k} \langle \phi_{u_t}^{u_{<t}} | \rho_{x,y} | \phi_{u_t}^{u_{<t}} \rangle \log\left(\langle \phi_{u_k}^{u_{<k}} | d\rho_{x,y} | \phi_{u_k}^{u_{<k}} \rangle\right)$$

$$= \frac{1}{M} \sum_{x,y,k} \sum_{u_1,\ldots,u_k} \prod_{t=1}^{k} \langle \phi_{u_t}^{u_{<t}} | \rho_{x,y} | \phi_{u_t}^{u_{<t}} \rangle (\langle \phi_{u_k}^{u_{<k}} | d\rho_{x,y} | \phi_{u_k}^{u_{<k}} \rangle - 1)$$

$$= \frac{1}{M} \sum_{x,y,k} \sum_{u_1,\ldots,u_k} \prod_{t=1}^{k} \langle \phi_{u_t}^{u_{<t}} | \rho_{x,y} | \phi_{u_t}^{u_{<t}} \rangle (\langle \phi_{u_k}^{u_{<k}} | d\rho_{x,y} | \phi_{u_k}^{u_{<k}} \rangle - 1)$$

$$= \frac{1}{M} \sum_{x,y,k} \sum_{u_1,\ldots,u_k} \prod_{t=1}^{k} \langle \phi_{u_t}^{u_{<t}} | \left( \frac{\mathbb{I}}{d} + \varepsilon \frac{O_{x,y}}{d} \right) | \phi_{u_t}^{u_{<t}} \rangle \langle \phi_{u_k}^{u_{<k}} | \varepsilon O_{x,y} | \phi_{u_k}^{u_{<k}} \rangle$$

$$= \frac{1}{M} \sum_{x,y,k} \sum_{u_1,\ldots,u_{k-1}} \prod_{t=1}^{k-1} \langle \phi_{u_t}^{u_{<t}} | \left( \frac{\mathbb{I}}{d} + \varepsilon \frac{O_{x,y}}{d} \right) | \phi_{u_t}^{u_{<t}} \rangle \sum_{u_k} \frac{1}{d} \langle \phi_{u_k}^{u_{<k}} | I + \varepsilon O_{x,y} | \phi_{u_k}^{u_{<k}} \rangle \langle \phi_{u_k}^{u_{<k}} | \varepsilon O_{x,y} | \phi_{u_k}^{u_{<k}} \rangle$$

$$= \frac{1}{M} \sum_{x,y,k} \sum_{u_1,\ldots,u_{k-1}} \prod_{t=1}^{k-1} \langle \phi_{u_t}^{u_{<t}} | \left( \frac{\mathbb{I}}{d} + \varepsilon \frac{O_{x,y}}{d} \right) | \phi_{u_t}^{u_{<t}} \rangle \sum_{u_k} \frac{1}{d} \langle \phi_{u_k}^{u_{<k}} | \varepsilon O_{x,y} | \phi_{u_k}^{u_{<k}} \rangle$$

$$+ \frac{1}{M} \sum_{x,y,k} \sum_{u_1,\ldots,u_{k-1}} \prod_{t=1}^{k-1} \langle \phi_{u_t}^{u_{<t}} | \left( \frac{\mathbb{I}}{d} + \varepsilon \frac{O_{x,y}}{d} \right) | \phi_{u_t}^{u_{<t}} \rangle \sum_{u_k} \frac{1}{d} \langle \phi_{u_k}^{u_{<k}} | \varepsilon O_{x,y} | \phi_{u_k}^{u_{<k}} \rangle^2$$

$$\leq \frac{1}{M} \sum_{x,y,k} \sum_{u_1,\ldots,u_{k-1}} \prod_{t=1}^{k-1} \langle \phi_{u_t}^{u_{<t}} | \left( \frac{\mathbb{I}}{d} + \varepsilon \frac{O_{x,y}}{d} \right) | \phi_{u_t}^{u_{<t}} \rangle \times \frac{1}{d} \times \text{tr}(\varepsilon O_{x,y})$$

$$+ \frac{1}{M} \sum_{x,y,k} \sum_{u_1,\ldots,u_{k-1}} \prod_{t=1}^{k-1} \langle \phi_{u_t}^{u_{<t}} | \left( \frac{\mathbb{I}}{d} + \varepsilon \frac{O_{x,y}}{d} \right) | \phi_{u_t}^{u_{<t}} \rangle \times \frac{1}{d} \times \text{tr}(\varepsilon^2 O_{x,y}^2)$$

$$\leq \frac{1}{M} \sum_{x,y,k} \sum_{u_1,\ldots,u_{k-1}} \prod_{t=1}^{k-1} \langle \phi_{u_t}^{u_{<t}} | \left( \frac{\mathbb{I}}{d} + \varepsilon \frac{O_{x,y}}{d} \right) | \phi_{u_t}^{u_{<t}} \rangle \times \frac{1}{d} \times 11^2 d\varepsilon^2$$

$$\leq \frac{1}{M} \sum_{x,y,k} 1 \times 11^2 \varepsilon^2$$

$$\leq 11^2 N \varepsilon^2,$$

where we use again $\sum_{u_t} \langle \phi_{u_t}^{u_{<t}} | O_{x,y} | \phi_{u_t}^{u_{<t}} \rangle = \text{tr}(O_{x,y}) = 0$ for all $t$ and

$$\sum_{u_k} \langle \phi_{u_k}^{u_{<k}} | O_{x,y} | \phi_{u_k}^{u_{<k}} \rangle^2 = \sum_{u_k} \text{tr}(O_{x,y} | \phi_{u_k}^{u_{<k}} \rangle \langle \phi_{u_k}^{u_{<k}} | O_{x,y} | \phi_{u_k}^{u_{<k}} \rangle \langle \phi_{u_k}^{u_{<k}} |)$$

$$\leq \sum_{u_k} \text{tr}(O_{x,y}^2 | \phi_{u_k}^{u_{<k}} \rangle \langle \phi_{u_k}^{u_{<k}} |) = \text{tr}(O_{x,y}^2) \leq 11^2 d.$$

Therefore the mutual information can be upper bounded by

$$I(X, Y : X', Y') \leq 121N\varepsilon^2 + 2N \sup_{\phi, \|\phi\|_2 \leq 1} \frac{1}{M} \sum_{|x|, y \leq m/2} \langle \phi | U_y O_{x,y} U_y^\dagger | \phi \rangle^2 \varepsilon^2$$

$$\leq 123N\varepsilon^2.$$

Since the mutual information is always lower bounded by $\Omega(\log(m) + d)$ we conclude that $N = \Omega((d + \log(m)/\varepsilon^2)$. Finally, we have proven the following lower bound on adaptive strategies for hypothesis selection problem:

**Proposition C.8.** *Any learning algorithm with adaptive incoherent measurements requires*

$$N \geq \Omega\left(\frac{d + \log(m)}{\varepsilon^2}\right)$$

*copies of $\rho$ to find the closest quantum state $\sigma_{i^\star}$ to $\rho$ and approximate $\rho$ to at most $\varepsilon/10$ with at least a probability $2/3$.*

This Proposition implies that Alg. 1 is almost optimal and $\tilde{\Theta}(d/\varepsilon^2)$ is the optimal copy complexity of hypothesis selection problem using adaptive incoherent measurements.

## D Technical lemmas

In this section we group technical lemmas useful for the previous proofs of this article.

### D.1 Weingarten Calculus

Since we use generally a uniform POVM, which consists in sampling a Haar-unitary matrix, we need some facts from Weingarten calculus in order to compute Haar-unitary intergrals. If $\pi$ a permutation of $[n]$, let $\mathrm{Wg}(\pi, d)$ denotes the Weingarten function of dimension $d$. The following lemma is crucial for our results.

**Lemma D.1.** *(Gu, 2013) Let $U$ be a $d \times d$ Haar-distributed unitary matrix and $\{A_i, B_i\}_i$ a sequence of $d \times d$ complex matrices. We have the following formula*

$$\mathbb{E}\left(\mathrm{tr}(UB_1U^*A_1U \ldots UB_nU^*A_n)\right) = \sum_{\alpha, \beta \in \mathcal{S}_n} \mathrm{Wg}(\beta\alpha^{-1}, d)\mathrm{tr}_{\beta^{-1}}(B_1, \ldots, B_n)\mathrm{tr}_{\alpha\gamma_n}(A_1, \ldots, A_n),$$

*where $\gamma_n = (12 \ldots n)$ and $\mathrm{tr}_\sigma(M_1, \ldots, M_n) = \Pi_j \mathrm{tr}(\Pi_{i \in C_j} M_i)$ for $\sigma = \Pi_j C_j$ and $C_j$ are cycles.*

We need also some values of Weingarten function:

**Lemma D.2.**
- $\mathrm{Wg}((1), d) = \frac{1}{d}$,
- $\mathrm{Wg}((12), d) = \frac{-1}{d(d^2-1)}$,
- $\mathrm{Wg}((1)(2), d) = \frac{1}{d^2-1}$,
- $\mathrm{Wg}((123), d) = \frac{2}{d(d^2-1)(d^2-4)}$,
- $\mathrm{Wg}((12)(3), d) = \frac{-1}{(d^2-1)(d^2-4)}$,
- $\mathrm{Wg}((1)(2)(3), d) = \frac{d^2-2}{d(d^2-1)(d^2-4)}$.

### D.2 Concentration inequalities for Haar-random unitary matrices

**Theorem D.3.** *(Meckes et al., 2013) Let $M = U(d)^k$ endowed by the $L_2$-norm of Hilbert-Schmidt metric. If $F : M \to \mathbb{R}$ is $L$-Lipschitz, then for any $t > 0$*

$$\mathbb{P}\left(|F(U_1, \ldots, U_k) - \mathbb{E}\left(F(U_1, \ldots, U_k)\right)| \geq t\right) \leq e^{-dt^2/12L^2},$$

*where $U_1, \ldots, U_k$ are independent Haar-distributed unitary matrices.*

