# OpenReview forum: "On Adaptivity in Quantum Testing"
_TMLR — Accepted by TMLR_

### Review · Reviewer_csnD · 2023-06-04

**Summary Of Contributions:**

This paper studies the problem of quantum hypothesis selection, with the focus on understanding whether using adaptive strategies can significantly improve the sample complexity of quantum states. Specifically, the paper proved that for testing between two states, there is a factor of 4 improvement compared to non-adaptive strategies. A more significant speedup is established for the learning with shadow problem in Huang et al. (2020), whose adaptive complexity is ~O(d/eps^2), but this paper proved that non-adaptively it takes Omega(d^2/eps^2) copies.

**Audience:**

Yes

**Broader Impact Concerns:**

I don't think this work has any broader impact concerns.

**Claims And Evidence:**

Yes

**Requested Changes:**

Please try to address my two concerns in weaknesses above.

In addition, it would be helpful if the authors can make improvements on the following points:

- Between consecutive paragraphs, there should be an empty line for spacing. The current template is a bit dense for reading.

- There are a few other papers about property testing of quantum states/distributions that the authors should probably cite, for instance Acharya et al. https://ieeexplore.ieee.org/abstract/document/9163139/ (copy complexity of estimating entropies of a quantum state), Gilyen and Li https://arxiv.org/abs/1902.00814 (it studied ell_1 and ell_2 norm closeness testing problems), and Belovs https://arxiv.org/pdf/1904.02192.pdf (distinguishing between classical distributions encoded by quantum oracles).

- I’m a bit confused by Section 3.1.2. It seems that the adaptive strategy still uses the optimal POVM by the Holevo-Helstrom theorem, so how can the bound improve by a factor of 4 compared to the non-adaptive case? More explanations will be helpful.

**Strengths And Weaknesses:**

Strengths: The influence of adaptivity is an interesting topic, and it reflects the difference between offline learning and online learning. It is nice to know that in the quantum setting, namely the hypothesis selection of quantum states, there’s provable separation between adaptive and non-adaptive strategies.

Weaknesses: I think this work has two main weaknesses: novelty and connection to machine learning.

In terms of novelty, the problem studied in Proposition 3.1, non-adaptive testing of quantum states, is called quantum state discrimination and is a standard problem in quantum information theory. Its adaptive version, Proposition 3.3, has significant overlap with Li et al. (2022b). In terms of the classical shadow problem studied in Section 4, Theorem 4.1 is from Huang et al. (2020). In general, my personal take on this paper is that the perspective of telling the story in adaptivity vs. non-adaptivity is nice, but technically the contribution is not that significant and it glues existing techniques together into this high-level picture.

Another notable weakness is the connection to machine learning. In general, the technical results in this paper may suit better in quantum information theory or theoretical computer science, for instance most papers cited by this one fall into those two categories. So why is this paper particularly suitable for TMLR? I think the authors should think about this question and write the paper in a more favorable way to machine learning audiences. For instance, how is the adaptivity vs. non-adaptivity story in this paper same or different compared to classical online machine learning vs. offline machine learning? Do the results in this paper have further implications in classical machine learning? Answers to those questions will be very helpful.

---

### Review · Reviewer_W2pY · 2023-06-17

**Summary Of Contributions:**

The submission presents an exploration of whether sequential strategies can outperform non-sequential ones in solving the hypothesis selection problem, which is an important problem in quantum learning theory. The main focus is to understand how many measurements are needed to select the hypothesis with certain probability and error tolerances under different regimes, including sequential/non-sequential adaptive/non-adaptive algorithms.

The authors show that there are some situations where sequential or adaptive strategies require fewer measurements than non-adaptive non-sequential ones. For the setting of two hypotheses, the paper shows that sequential strategies outperform non-sequential ones by a factor of 4, which was also shown in Li et al. (2022b). For the setting of more hypotheses, the paper constructs a particular problem and establishes a provable polynomial separation between adaptive and non-adaptive strategies. Overall, this paper presents new insights into the performance of testing strategies for quantum systems, with potential implications for understanding the power and differences between adaptive and non-adaptive protocols in quantum information processing.

**Audience:**

Yes

**Broader Impact Concerns:**

The contribution of this paper is theoretical and I do not see any concerns.

**Claims And Evidence:**

Yes

**Requested Changes:**

Please refer to the comments on Weaknesses.

**Strengths And Weaknesses:**

### Strengths
1. The paper presents solid theoretical results for analyzing the performance of hypothesis testing strategies in quantum learning theory and rigorously establishes a polynomial separation between adaptive and non-adaptive strategies.
2. The technical contributions made by this paper in analyzing the adaptive and non-adaptive strategies for hypothesis selection are novel, advancing our understanding of the fundamental question of the advantages of the adaptive regime.
3. The results and insights presented in the paper have potential implications for quantum information theory.

### Weaknesses
1. The introduction could be improved by more clearly explaining the motivation for this problem, and the significance of the research question being addressed. As for this journal TMLR, it will be better to mention more background/motivations on how the problem is related to (quantum) learning theory.
2. The paper could benefit from a more systematic organization of its main results, including clear definitions and a table of the different settings under consideration. While reading, the current versions sometimes confused me about these many different settings and terminologies. The clear presentation of the definition and conclusion could be improved to make the results more accessible to a wider range of readers.
3. One main result in this paper is the advantage established in section 4. It is unclear to me about the intuition behind the constructed problem and meanwhile it seems not very direct that how the constructed problem is a hypothesis selection problem. Also, are there any further applications of this problem in quantum information or learning theory？

---

### Review · Reviewer_9Wc6 · 2023-06-18

**Summary Of Contributions:**

**Disclaimer**: In all fairness to the authors, I do not know anything about quantum testing / quantum theory, and my review here is cosmetic at best going through just the organizational aspects of the writeup as suggested by the esteemed Action Editor. The organizational aspects of the paper appear fine.

This paper considers the quantum hypothesis selection problem. The authors characterize and compare the sample complexity for sequential and non-sequential strategies showing that sequential adaptive strategies are more efficient than non-sequential non-adaptive ones for some specific problems.



**Audience:**

Yes

**Claims And Evidence:**

Yes

**Requested Changes:**

NA

**Strengths And Weaknesses:**

The authors characterize and compare the sample complexity for sequential and non-sequential strategies showing that sequential adaptive strategies are more efficient than non-sequential non-adaptive ones for some specific problems.

---

### Decision · Action_Editors · 2023-08-14

**Recommendation:** Accept as is

**Comment:**

The authors study whether sequential strategies can outperform non-sequential ones in solving the hypothesis selection problem, which is an important problem in quantum learning theory. The main focus is to understand how many measurements are needed to select the hypothesis with certain probability and error tolerances under different regimes, including sequential/non-sequential adaptive/non-adaptive algorithms.

The authors show that there are some situations where sequential or adaptive strategies require fewer measurements than non-adaptive non-sequential ones. For the setting of two hypotheses, the paper shows that sequential strategies outperform non-sequential ones by a factor of 4, which was also shown in Li et al. (2022b). For the setting of more hypotheses, the paper constructs a particular problem and establishes a provable polynomial separation between adaptive and non-adaptive strategies.

Overall, this paper presents new insights into the performance of testing strategies for quantum systems, with potential implications for understanding the power and differences between adaptive and non-adaptive protocols in quantum information processing.

**Audience:**

A very narrow sub-community of ML interested in quantum testing

**Claims And Evidence:**

Key claims are supported by proofs.